# Combating Dual Noise Effect in Spatial-temporal Forecasting via Information Bottleneck Principle

## Abstract

Spatial-temporal forecasting plays a pivotal role in urban planning and computing. Although Spatial-Temporal Graph Neural Networks (STGNNs) excel in modeling spatial-temporal dynamics, they often suffer from relatively poor computational efficiency. Recently, Multi-Layer Perceptrons (MLPs) have gained popularity in spatial-temporal forecasting for their simplified architecture and better efficiency. However, existing MLP-based models can be susceptible to noise interference, especially when the noise can affect both input and target sequences in spatial-temporal forecasting on noisy data. To alleviate this impact, we propose *Robust Spatial-Temporal Information Bottleneck (RSTIB)* principle. The RSTIB extends previous Information Bottleneck (IB) approaches by lifting the specific Markov assumption without impairing the IB nature. Then, by explicitly minimizing the irrelevant noisy information, the representation learning guided by RSTIB can be more robust against noise interference. Furthermore, the instantiation, RSTIB-MLP, can be seamlessly implemented with MLPs, thereby achieving efficient and robust spatial-temporal modeling. Moreover, a training regime is designed to handle the dynamic nature of spatial-temporal relationships by incorporating a knowledge distillation module to alleviate feature collapse and enhance model robustness under noisy conditions. Our extensive experimental results on six intrinsically noisy benchmark datasets from various domains show that the RSTIB-MLP runs much faster than state-of-the-art STGNNs and delivers superior forecasting accuracy across noisy environments, substantiating its robustness and efficiency.

## 1 Introduction

Spatial-temporal forecasting holds great significance in modeling complex dynamic systems (Bai et al., 2020; Guo et al., 2021a; Deng et al., 2021). It needs to capture both temporal and spatial dependencies to accurately predict important statistics, *e.g.*, traffic flow states or electricity consumption.

Previous works in spatial-temporal forecasting have effectively adopted convolutional neural networks (CNNs) (Lai et al., 2018), recurrent neural networks (RNNs) (Meng et al., 2020), temporal convolution networks (TCNs) (Wu et al., 2019) to model spatial-temporal relations. Lately, there has been a growing interest in spatial-temporal graph neural networks (STGNNs) (Shao et al., 2022b;c; Wu et al., 2019) due to their strong capacity. Though achieving exceptional performance, STGNN-based methods suffer from slow computational efficiency. To alleviate this, a few recent works (Shao et al., 2022a; Qin et al., 2023; Wang et al., 2023b; Yi et al., 2024) adopt Multi-Layer Perceptrons (MLP) due to its advantageous efficiency. However, MLP-based models become less effective when facing spatial-temporal noise perturbation, which is common in real world (Jiang et al., 2023b; Tang et al., 2023; Fang et al., 2021; Liu et al., 2024c; Zhang et al., 2023). As shown in Fig. 1, two time series may become indistinguishable in both the historical input end and the forecasting target end due to the presence of noise perturbation. This is termed as "*sample indistinguishability*" in (Shao et al., 2022a). There is no theoretically grounded principle in existing MLP-based models that can alleviate this issue. Besides, we can also observe severe feature collapse, *i.e.*, feature collapse is reflected by much lower feature variance, which is used for quantitative analysis of the diversity among the learned features (Papyan et al., 2020; Zhu et al., 2023a; Bardes et al., 2021) (See Section 3).

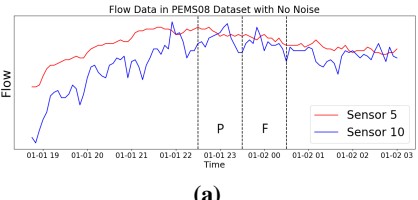 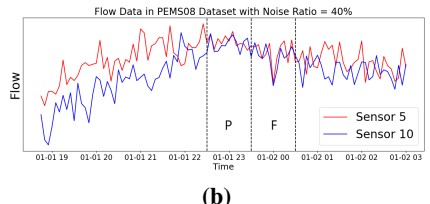

**(a)**                     **(b)**

Figure 1: **(a) Two time series are distinguishable at both historical input (P area) and forecasting target (F area), but (b) they become indistinguishable in both cases in the presence of noise.**

Methods based on adversarial training (Jiang et al., 2023b), graph information bottleneck (GIB) (Tang et al., 2023), mathematical tools (Choi et al., 2022), frequency domain MLPs (Yi et al., 2024), Biased TCN (Chen et al., 2024), Spatial-temporal Curriculum Dropout (Wang et al., 2023a) have been proposed to combat the noise for robust representation learning. However, spatial-temporal data often undergoes preprocessing through a sliding window mechanism, where a sequence can serve as either the input or the target when residing in different windows. Noise potentially harms both ends, termed as "*dual noise effect*". Consequently, the enhancement of these methods is often marginal since they only consider a single end effect. While Robust Graph Information Bottleneck (RGIB) (Zhou et al., 2023) effectively combats bilateral edge noise for link prediction, we reveal that generalizing RGIB directly to MLP networks for spatial-temporal forecasting is difficult: the GNN model architecture and graph data assumption it relies upon is very different from the spatial-temporal data that is continuous multivariate time series. It is significant to derive additional guiding principles and specific instantiation to handle such a scenario.

In this paper, we first disclose that spatial-temporal data noise is detrimental to (i) both forecasting input and target and (ii) both predictive performance and feature variance. To combat it, we introduce a new theoretically sound principle, named *Robust Spatial-Temporal Information Bottleneck (RSTIB)*, generalizing the RGIB principle to mitigate the dual noise effect in spatial-temporal data. Particularly, it lifts the Markov assumption typically assumed in IB while not impairing the IB nature. In doing so, the derived additional noisy information and the original redundant information are explicitly reformulated and minimized. RSTIB-MLP, guided by the RSTIB principle, is further instantiated for robust spatial-temporal modeling. Subsequently, combined with the instantiation, we propose a training regime to handle the dynamic relations between different time series via an innovative knowledge distillation module. The key idea is to balance the informative terms within the objective by accounting for the quantified noise impact, thereby being better balanced and less impacted by noisy information. We quantify the noise impact to each time series by defining a new noise impact indicator (**Definition 4.2**) and incorporate this knowledge for each time series.

Our main contributions can be summarized as follows:

- To the best of our knowledge, it is the first work to derive and extend IB for handling the dual noise effect in spatial-temporal forecasting. We reveal that dual noise effect can lead to significant degradation in both of the predictive accuracy and feature variance.

- We propose the RSTIB principle, a general theoretical framework to robustify MLP networks. A corresponding computationally efficient instantiation, named RSTIB-MLP, is devised by utilizing pure MLP networks for robust forecasting on noisy spatial temporal data, with theoretical support for its robustness due to the RSTIB principle.

- A new training regime is further designed to enhance the robust representation learning. This regime incorporates a novel knowledge distillation module, strengthening robustness and boosting the variance of its learned features.

- Benchmark datasets from various domains are adopted for noisy and clean evaluations. Extensive comparisons based on our theoretical analysis and empirical studies demonstrate our method's superiority.

## 2 RELATED WORK

**Spatial-Temporal Forecasting (STF).** Efforts in STF have led to the development of sophisticated models such as AGCRN (Bai et al., 2020), GraphWaveNet (Wu et al., 2019), and STExplainer (Tang et al., 2023), which leverage STGNN-based methodologies to model series-wise dependencies over time. Recent explorations have integrated Neural-ODE-based (Jin et al., 2022) and self-supervised learning paradigms (Li et al., 2022) to enhance spatial-temporal modeling. Despite their predictive capabilities, these methods often suffer from computational efficiency issues when compared with MLP-based approaches.

**MLP-based Approaches for STF.** In response to the efficiency challenge, MLP-based approaches have gained attention. Notable works include STID (Shao et al., 2022a), which incorporates spatial-temporal identity information to achieve superior performance over STGNN-based methods, and STHMLP (Qin et al., 2023), which employs a hierarchical MLP structure to capture various aspects of spatial-temporal data. FreTS (Yi et al., 2024) applies MLPs in the frequency domain. Specifically, its advantage of the energy compaction can help MLPs to preserve clearer patterns while filtering out influence of noises. However, these methods have not yet explored the dual noise effect in the face of comprehensive noise perturbations.

**Robust Representation Learning with Information Bottleneck Principle.** The Information Bottleneck (IB) principle has emerged as a guiding framework for robust representation learning. Initially applied in Deep Variational Information Bottleneck (DVIB) (Alemi et al., 2016), IB has since found applications in diverse domains (Peng et al., 2018; Higgins et al., 2016). Notably, GIB (Wu et al., 2020) extends IB to graph-structured data for supervised learning. Subsequent advancements, such as STExplainer (Tang et al., 2023), build upon the GIB principle for explainable representations. While these methods can enhance robustness to some extent, they overlook the presence of noise in the forecasting target. RGIB (Zhou et al., 2023) takes a step forward by decoupling mutual information to enhance such robustness, but generalizing it to MLP networks for spatial-temporal forecasting remains unexplored.

## 3 NOTATIONS AND PRELIMINARIES

**Spatial-temporal Forecasting (STF).** STF aims at predicting the future target spatial-temporal data $Y \in \mathbb{R}^{F \times N \times C}$ with $N$ time series of $C$ features in each time series within $F$ nearest future time slots, based on historical input data $X^h \in \mathbb{R}^{P \times N \times C}$ from the past $P$ time slots. Additionally, we denote the sample from time series $i$ at time step $t$ as $X_{t,i}^h \in \mathbb{R}^C$ and $Y_{t,i} \in \mathbb{R}^C$ for the historical and future data respectively.

**Feature Variance.** Drawing inspiration from prior studies (Bardes et al., 2021; Zhu et al., 2023a), we aim for the learned representations in spatial-temporal forecasting to display significant diversity, capturing complex spatial-temporal patterns effectively. We quantify feature variance as follows:

Consider a set of latent spatial-temporal representations $(z_1, z_2, \ldots, z_N)$, where each $z_i \in \mathbb{R}^d$ for $i = 1, \ldots, N$. The feature variance is defined as:

$$\mathbf{Var}(z_1, z_2, \ldots, z_N) = \frac{1}{d} \sum_{i=1}^{d} \left( \sqrt{\mathbf{Cov}_{ii}} \right), \tag{1}$$

where $\mathbf{Cov}_{ii}$ denotes the variance of the $i$-th feature across the set of representations, defined as the diagonal elements of the covariance matrix $\mathbf{Cov}$. $\mathbf{Cov}$ is computed as $\mathbf{Cov} = \frac{1}{N-1} \sum_{i=1}^{N} (z_i - \bar{z})(z_i - \bar{z})^T$, with $\bar{z} = \frac{1}{N} \sum_{i=1}^{N} z_i$ representing the mean vector of the representations (see (Bardes et al., 2021; Zhu et al., 2023a) for detailed derivation and theoretical grounding).

**Noise Perturbation vs. Feature Variance.** We conduct an empirical study to assess the impact of noise perturbation on feature variance, where the STID model (Shao et al., 2022a) is used. During training, we inject random noise into the signals from the single input end and both ends, with a 50% probability across varying noise ratios – 10%, 30%, 50%, 70%, and 90%. The evaluation focuses on the diversity of extracted features by measuring the variance under these conditions. As shown in Table 1, a significant degradation in feature variance is observed with increasing noise perturbation. Besides, a faster degradation can be observed when injecting to both ends, highlighting the detrimental effects of the noise and the dual noise effect on the effectiveness of capturing spatial-temporal patterns.

**Table 1: Feature Variance of Single-End and Dual Noise Effect Under Different Noise Ratios**

| Noise Ratio | 10% | 30% | 50% | 70% | 90% |
|---|---|---|---|---|---|
| **Feature Variance (Single End)** | 1.9462 | 1.0325 | 0.8529 | 0.6323 | 0.6042 |
| **Feature Variance (Dual Noise Effect)** | 1.6900 | 0.6582 | 0.4859 | 0.4350 | 0.4332 |

## 4 METHODOLOGY

In this section, we introduce the Robust Spatial-Temporal Information Bottleneck (RSTIB) principle, a theoretical framework designed to be more general for enhancing robust spatial-temporal modeling. Following this, we detail a novel instantiation termed RSTIB-MLP, which leverages data reparameterization techniques for continuous multivariate time series. We also design a training regime, incorporating a knowledge distillation module, to further enhance the performance of spatial-temporal forecasting. This approach capitalizes on the dynamic spatial-temporal relationships inherent in the data, resulting in a better balance of informative terms within the objective.

### 4.1 DERIVING THE RSTIB PRINCIPLE

Let $X$ represents the input to the IB model and its variants, obtained from $X^h$ and the attachment of spatial-temporal information from a specially designed module. Formally, given the input $X$, target $Y$, and encoding $Z$ from $X$, the learning objective of the standard IB principle can be formulated as follows:

$$\min \mathcal{L}_{IB} = -I(Z, Y) + \beta \times I(X, Z), \tag{2}$$

where $I(\cdot, \cdot)$ denotes mutual information (MI), and $\beta \geq 0$ is a Lagrange multiplier for controlling the trade-off between the compression of $X$ and the preservation of $Y$. The Markov chain $Z - X - Y$ is assumed in IB (Alemi et al., 2016). We can use the information diagram (Fig.2a) to depict the IB, where we represent information of $X$ and $Y$ as circles. Then IB encourages to cover as much of $I(X;Y)$ and as little of $H(X|Y)$ as possible.

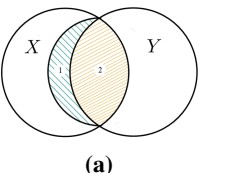
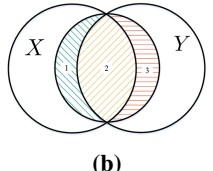

(a)                    (b)

**Figure 2: Comparison of IB(a) and DVIB with lifted Markov assumption $Z - X - Y$(b). (a) (1)** $H(X|Y)$ **information** $Z$ **covers,** *i.e.*, $I(X;Z|Y)$, **(2) the minimum sufficient information preserved by the expected optimal representation** $Z$, *i.e.*, $I(X;Y)$ **(b) By lifting** $Z - X - Y$, $I(Z;Y|X)$ **exists as (3),** *i.e.*, $H(Y|X)$ **information** $Z$ **covers.**

However, the vanilla IB is sub-optimal in our scenario. By drawing inspirations from (Jiang et al., 2023b; Choi et al., 2022; Tang et al., 2024; Yuan et al., 2024; Liu et al., 2024b), we firstly have the following assumptions about spatial-temporal data:

**Assumption 4.1.** *Noisy Nature of spatial-temporal Data. We focus on spatial-temporal data that inherently exhibits noisy characteristics. Under the sliding window mechanism, a sequence can serve different purposes when residing in different windows, either as the input or the target. Consequently, the noise elements can potentially reside in the input and target areas. For simplicity, we presume the noise type in our analysis to adhere to Additive White Gaussian Noise (AWGN), a prevalent and empirically approximated noise model in practical applications (Lim & Puthusserypady, 2007).*

**Assumption 4.2.** *Invariant and Variant spatial-temporal Patterns. A dynamic spatial-temporal graph exhibits a dual nature, wherein each node, representing a time series, embodies both spatial-temporal invariant patterns conducive to generalized predictions across all time windows and spatial-temporal variant patterns reflecting underlying time-varying and node-specific dynamics.*

Following these assumptions, it is naive to assume the Markov assumption $Z - X - Y$, which results in $I(Z; Y \mid X) = 0$. This implies that we directly overlook the noisy information behind

$H(Y|X)$ (*i.e.*, the noisy information conveyed by the target data). Fortunately, Wieczorek & Roth (2020) demonstrate that a lower bound of $I(Z;Y)$ can be derived without relying on the $Z - X - Y$ assumption, thus lifting the $Z - X - Y$ Markov restriction. It is achieved in the DVIB model, which assumes $X - Z - Y$ assumption by its construction. We apply it to our specific scenario by assuming only the Markov chain condition $X - Z - Y$ (See **Proposition 4.5**). The introduced additional term, $I(Z;Y \mid X)$, as represented in Fig.2b, must be minimized as well, along with the original irrelevant information, *i.e.*, $I(X;Z|Y)$. Accordingly, we introduce the following reformulations:

**Proposition 4.1.** *Reformulate* $I(Z;Y \mid X)$ *and* $I(X;Z \mid Y)$. *The sum of* $I(Z;Y \mid X)$ *and* $I(Y;Z|X)$ *can be reformulated as:* $I(Z;Y \mid X) + I(Z;X \mid Y) = I(Z;X,Y) - I(X;Y;Z)$. *Proof. See Appendix F.1.*

Leveraging this reformulation, we aim to minimize the influence of noisy information Z captures, encapsulated by $H(X|Y)$ and $H(Y|X)$.

**Definition 4.1.** *Robust Spatial-Temporal Information Bottleneck Principle.* *Under the Markov chain condition* $X - Z - Y$, *the learning objective is encapsulated by the following optimization:*

$$\min \mathcal{L}_{RSTIB} = -I(Z,Y) + \beta_1 \times I(Z;X,Y) - \beta_2 \times I(X;Y;Z). \tag{3}$$

*where* $\beta_1, \beta_2 \geq 0$ *is the respective Lagrange multipliers to control the balance within this objective.*

### 4.2 INSTANTIATING RSTIB

Here, we introduce the instantiation, termed RSTIB-MLP, in the order of $I(X;Y;Z)$, $I(Z;X,Y)$ and $I(Z,Y)$.

**Instantiating** $I(X;Y;Z)$. Per definition (**Definition B.8**), the expression $I(X;Y;Z) = I(X;Y) - I(X;Y \mid Z)$ indicates that maximizing $I(X;Y;Z)$ is equivalent to minimizing $I(X;Y \mid Z)$, given $I(X;Y)$ remains constant. It is important to note that the Markov chain condition $X - Z - Y$ is only approximated by reaching the optimal joint distribution of $X, Y, Z$. Therefore, by explicitly minimizing $I(X;Y \mid Z)$, we aim to learn a sufficient $Z$ while reaching our objective simultaneously. To this end, $Z$ is initially encoded from X. Then, we aim to reduce the relative knowledge between $X$ and $Y$ by observing $Z$. To achieve this objective, we employ data reparameterization to obtain the reparameterized $\tilde{X}$ and $\tilde{Y}$ while assuming independent and identically distributed (i.i.d) prior distributions of them, thereby reducing the overlapped information conditioned on $X, Z$, and $Y$. We effectuate the instantiation by directly imposing input regularization $I(\tilde{X};X)$ and target regularization $I(\tilde{Y};Y)$. While mutual information terms are typically intractable, we introduce upper bounds for $I(\tilde{X};X)$ and $I(\tilde{Y};Y)$, as elucidated in **Proposition 4.2**.

**Proposition 4.2.** *The Upper Bounds of* $I(\tilde{X};X)$ *and* $I(\tilde{Y};Y)$. *Assuming the prior distribution of* $\tilde{X}$ *and* $\tilde{Y}$, *denoted as* $Q(\tilde{X})$ *and* $Q(\tilde{Y})$, *to be i.i.d unit Gaussian* $\mathcal{N}(0,1)$. *The upper bounds for* $I(\tilde{X};X)$ *and* $I(\tilde{Y};Y)$ *are given by* $I(\tilde{X};X) \leq \mathbb{E}\left[KL\left(P_{\phi_x}(\tilde{X}|X)||Q(\tilde{X})\right)\right]$ *and* $I(\tilde{Y};Y) \leq \mathbb{E}\left[KL\left(P_{\phi_y}(\tilde{Y}|Y)||Q(\tilde{Y})\right)\right]$, *where* $KL$ *denotes the Kullback–Leibler divergence,* $P_{\phi_x}$ *and* $P_{\phi_y}$ *denote the parameterized distributions. Proof. See Appendix F.2.*

According to **Proposition 4.2**, we first utilize simple MLP layers to parameterize the posterior distribution $P_{\phi_z}(Z|X)$. This parameterization yields the posterior Gaussian distribution of $Z$, represented as $P_{\phi_z} \sim \mathcal{N}(\mu_z, \sigma_z^2)$. Subsequently, we employ two additional Fully-Connected(FC) layers, one for $X$ and the other for $Y$, to facilitate dimension transformation for aligning the dimensions of $X$ and $Y$ respectively. This process parameterizes two distributions, denoted as $P_{\hat{\phi}_x} \sim \mathcal{N}(\hat{\mu_x}, \hat{\sigma_x}^2)$ and $P_{\hat{\phi}_y} \sim \mathcal{N}(\hat{\mu_y}, \hat{\sigma_y}^2)$. According to this, we establish $P_{\phi_x} \sim \mathcal{N}(\mu_x, \sigma_x^2)$ and $P_{\phi_y} \sim \mathcal{N}(\mu_y, \sigma_y^2)$, where $\mu_x = x + \hat{\mu_x}$, $\mu_y = y + \hat{\mu_y}$, $\sigma_x^2 = \hat{\sigma_x}^2$ and $\sigma_y^2 = \hat{\sigma_y}^2$ respectively. Then, we adopt data reparameterization to obtain $\tilde{x} = \mu_x + \sigma_x \epsilon$ and $\tilde{y} = \mu_y + \sigma_y \epsilon$, where $\tilde{x}$ and $\tilde{y}$ represent the reparameterized signals, with each $\tilde{x} \in \tilde{X}$ and $\tilde{y} \in \tilde{Y}$ respectively, and $\epsilon \sim \mathcal{N}(0,1)$.

**Proposition 4.3.** *Analytical Solution for the Upper Bounds of the Input and Target Regularization.* *The Kullback-Leibler (KL) divergence between two Gaussian distributions, given their means and variances, can be analytically determined. Specifically, in our context, the KL divergence is computed for the input and target respectively, as* $\mathcal{L}_x = KL\big(\mathcal{N}(\mu_x, \sigma_x^2)||\mathcal{N}(0,1)\big) =$

$\frac{1}{2}\left(-\log \sigma_x^2 + \mu_x^2 + \sigma_x^2 - 1\right)$, $\mathcal{L}_y = KL\left(\mathcal{N}(\mu_y, \sigma_y^2)\,\|\,\mathcal{N}(0,1)\right) = \frac{1}{2}\left(-\log \sigma_y^2 + \mu_y^2 + \sigma_y^2 - 1\right)$, *where $\mathcal{L}_x$ denotes the upper bound of the input regularization, and $\mathcal{L}_y$ denotes the upper bound of the target regularization. Proof. See Appendix F.3.*

**Instantiating** $I(Z; X, Y)$**.** According to the mutual information w.r.t three random variables (**Definition B.3**), we can express $I(Z; X, Y) = H(Z) - H(Z \mid X, Y)$. Thus, our objective is to minimize the overlap of the entropy of $Z$ with respect to $X$ and $Y$. Given the condition of the Markov chain $X - Z - Y$, this can be implemented by reducing the entropy overlap between $Z$ and $X$ through the use of data reparameterization. Specifically, reparameterized data $\tilde{X}$ serves as the input to the MLP encoders. The encoders maintain the same network structure and parameters as used in the instantiation of $I(X; Y; Z)$. Then, the posterior distribution $P_{\phi_z}(Z|\tilde{X})$ is parameterized through the encoding process, as denoted $P_{\phi_z} \sim \mathcal{N}(\mu_z, \sigma_z^2)$. The encoding $z = \mu_z + \sigma_z \epsilon$ is obtained through reparameterization, where $z \in Z$. Besides, we impose the representation regularization $I(\tilde{X}; Z)$, and by assuming the prior distribution of $Z$ to be similar i.i.d unit Gaussian $\mathcal{N}(0, 1)$, out goal can be reached. The analytical solution for the upper bound of $I(\tilde{X}; Z)$ is also given in **Proposition 4.4**.

**Proposition 4.4.** *The Upper Bound of the Representation Regularization and Its Analytical Solution. The Upper bound of the representation regularization can be similarly given by $I(\tilde{X}; Z) \leq \mathbb{E}\left[KL\left(P_{\phi_z}(Z|\tilde{X})\|Q(Z)\right)\right]$, with $Q(Z)$ being an i.i.d unit Gaussian $\mathcal{N}(0, 1)$. Specifically, we have the analytical solution for this upper bound: $\mathcal{L}_z = KL\left(\mathcal{N}(\mu_z, \sigma_z^2)\,\|\,\mathcal{N}(0,1)\right) = \frac{1}{2}\left(-\log \sigma_z^2 + \mu_z^2 + \sigma_z^2 - 1\right)$, where $\mathcal{L}_z$ denotes the upper bound of representation regularization. Proof. See Appendix F.4.*

**Instantiating** $I(Z; Y)$**.** Given the explicit reparameterization of $Y$ to obtain $\tilde{Y}$, we aim to optimize $I(Z; \tilde{Y})$ instead of $I(Z; Y)$. However, directly computing $I(Z; \tilde{Y})$ is also intractable. Therefore, we introduce **Proposition 4.5** below to provide an approximated lower bound.

**Proposition 4.5.** *The Lower Bound of $I(Z; \tilde{Y})$. The variational lower bound of $I(Z; \tilde{Y})$ can be derived and approximated by minimizing the typical regression loss while without being restricted to the Markov assumption $Z - X - Y$, as follows:*

$$I(Z; \tilde{Y}) \geq \mathbb{E}_{P(X)}\mathbb{E}_{P(Z|X)P(\tilde{Y}|X)} \log Q(\tilde{Y}|Z) \approx -\mathcal{L}_{reg}(Y^S, \tilde{Y}), \tag{4}$$

*where $\mathcal{L}_{reg}$ represents the regression loss and $Y^S$ signifies the prediction. Proof. See Appendix F.5.*

Specifically, predictions are made through a regression layer to obtain $Y^S$ based on $Z$. We employ a standard regression loss, such as Mean Absolute Error (MAE), to maximize the variational lower bound.

Furthermore, findings from (Burgess et al., 2018) underscore the significance of the training regime concerning the $\beta$ hyperparameter value for robust representation learning rather than adhering to a fixed $\beta$-weighted term. In light of Assumption 4.2, it is evident that conventional IB methods designed for static relations are not directly applicable to the domain of spatial-temporal forecasting, which inherently relies on dynamic relationships. Consequently, we adopt a novel approach by designing a training regime tailored for dynamic relations.

**Training Regime.** In our training regime, we adapt the regularization strategy (*i.e.*, the balance of the informative terms within the objective) to accommodate the noise impact on different time series quantified in each time window. When noise impact is low, we relax the regularization. When there is a significant noise impact, we intensify the regularization. To quantify it, we leverage a trained model with no assumption on the model type and treat it as the teacher. Then, the noise impact indicators, defined in **Definition 4.2**, are computed based on the teacher model's predictive performance. By leveraging this knowledge, we dynamically balance the RSTIB-MLP's robust representation learning in different time series within different time windows. The noise impact indicator is formally defined as follows:

**Definition 4.2.** *Noise Impact Indicator. Given the historical data $X^h \in \mathbb{R}^{T \times N \times C}$ and a teacher model $f_T$ with trained and fixed parameters, we define the noise impact indicator to quantify the*

*noise impact on each time series. It is calculated as follows:*

$$\hat{\alpha}_i = \frac{\exp\left(D\left(Y_i^T, Y_i\right)\right)}{\sum_{j=1}^{N} \exp\left(D\left(Y_j^T, Y_j\right)\right)} = \frac{\exp\left(D\left(f_T(A, X^h)_i, Y_i\right)\right)}{\sum_{j=1}^{N} \exp\left(D\left(f_T(A, X^h)_j, Y_j\right)\right)}, \qquad \forall i \in \{1, \ldots, N\}, \quad (5)$$

*where $A \in \mathbb{R}^{N \times N}$ represents the adjacency matrix, utilized optionally depending on the modeling approach of the teacher. $D(\cdot, \cdot)$ denotes the distance function, such as mean squared error (MSE) or mean absolute error (MAE), to indicate the predictive performance. The computed $\hat{\alpha}_i$ for each time series reflects the relative impact of noise within the current time window, with higher values indicating greater susceptibility to noise.*

**Learning Framework.** The final objective for the robust representation learning in RSTIB-MLP is formalized as follows:

$$\mathcal{L}_{RSTIB-MLP} = \sum_{i=1}^{N} \left[ -\mathcal{L}_{\text{reg}}(Y_i^S, \tilde{Y}_i) + (1 + \hat{\alpha}_i)(\lambda_x \mathcal{L}_{x,i} + \lambda_y \mathcal{L}_{y,i} + \lambda_z \mathcal{L}_{z,i}) \right]. \quad (6)$$

The balance among all terms is controlled by the noise impact indicator $\hat{\alpha}_i$ and the Lagrange multipliers $\lambda_x$, $\lambda_y$, and $\lambda_z$ for input, target, and representation regularization, respectively. This learning objective highlights the relationship between RSTIB-MLP and the proposed training regime. The control over the balance of the informative terms is achieved not only by setting the hyperparameters, namely the Lagrange multipliers, but also by leveraging knowledge from noise impact indicators computed for different time series.

## 5 EXPERIMENTS

**Datasets.** For demonstrating universality, we consider six datasets from different domains, including **PEMS04**, **PEMS07**, **PEMS08** (Fang et al., 2021; Guo et al., 2019; Song et al., 2020; Yu et al., 2017), **LargeST(SD)** (Liu et al., 2024a), **Weather2K-R** (Zhu et al., 2023b), **Electricity** (Deng et al., 2021). The diverse sample rates ensure the exploration of short-term, mid-term and long-term forecasting evaluations. Detailed statistics and public accesses are provided in *Appendix.*E. For **PEMS** and **LargeST(SD)** benchmark datasets, we choose the **traffic flow** (vehicles per hour) as the metric. For **Weather2K-R** dataset, We select **vertical visibility** from 20 meteorological factors as the experimental variable. For **Electricity** dataset, we select the **average electricity consumption** (Deng et al., 2021). Besides, For **Electricity** dataset, we adopt the same training, validation, and testing split ratio as in (Deng et al., 2021), and for other datasets, we adopt **6:2:2** for all datasets to ensure consistency.

**Robust Baselines for Clean and Noisy Spatial-temporal Forecasting.** (1) **MLP-based Baseline**: STID (Shao et al., 2022a); (2) **STGNN-based Methods**: GWN (Wu et al., 2019) (3) **IB-based Method**: STGKD (Tang et al., 2024) (4) **GIB-based Baselines**: STExplainer (Tang et al., 2023) and STExplainer-CGIB (STExplainer with Conventional GIB); (5) **Adversarial Training-based Method**: TrendGCN (Jiang et al., 2023b) (6) **Mathematical Tools-based Method**: STG-NCDE (Choi et al., 2022). (7) **Energy Compaction Enhanced Method** : FreTS (Yi et al., 2024) (8) **Biased TCN-based Method**: BiTGraph (Chen et al., 2024). (9) **Spatial-temporal Curriculum Learning-based Method**: STC-Dropout (Wang et al., 2023a).

**Extra Baselines Designed for Clean Spatial-temporal Traffic Forecasting.** We also dedicate to utilize PEMS datasets to compare RSTIB-MLP with three types of baselines proposed for clean spatial-temporal traffic forecasting: (1) **Attention-based Method**: DSTAGNN (Lan et al., 2022); (2) **MLP-based method**: STHMLP (Qin et al., 2023); (3) **STGNN-based Methods**: STGCN (Yu et al., 2017), AGCRN (Bai et al., 2020), GMSDR (Liu et al., 2022), FOGS (Rao et al., 2022), and TrendGCN (Jiang et al., 2023b);

**Implementation Details.** For the basic settings, we employ a hidden dimension $d = 64$ and utilize an MLP architecture with L=3 layers. For **PEMS** and **LargeTS(SD)** benchmark datasets, we use historical traffic flow data with window length P = 12 to forecast future traffic flow data with window length F = 12, while for the **Electricity** dataset, we follow the default settings in (Deng et al., 2021), *i.e.*, we set P=16 and F=3, and calculate the average predictive accuracy by averaging over 1, 2, 3 hours. Since there is no pre-defined graph structure in the Electricity dataset, some results are

denoted as "-" , meaning *Not Available*. The model performance is evaluated using three metrics: MAE, RMSE, and MAPE. The learning rate is initialized as $\eta = 0.002$ with a decay factor $r = 0.5$. Baselines with recommended hyperparameter settings are used (See *Appendix.*D). Our method is teacher model agnostic (*Appendix.*K.10), where we set the default teacher model to STGCN. Spatial-temporal prompts (Tang et al., 2024) are utilized to attach the spatial-temporal information.

## 5.1 MAIN RESULTS

**Table 2: Predictive Accuracy Comparison Under Various Noise Ratios in Different Datasets**

| Noise Ratio | 0%(clean) | | | 10% | | | 30% | | | 50% | | |
|---|---|---|---|---|---|---|---|---|---|---|---|---|
| Metrics | MAE | RMSE | MAPE | MAE | RMSE | MAPE | MAE | RMSE | MAPE | MAE | RMSE | MAPE |
| Dataset | PEMS04 | | | | | | | | | | | |
| STID | 18.79 | 30.37 | 12.51% | 27.83 | 41.34 | 17.31% | 36.53 | 52.74 | 21.11% | 36.22 | 52.15 | 21.45% |
| GWN | 19.22 | 30.74 | 12.52% | 30.03 | 43.27 | 19.32% | 39.55 | 56.78 | 22.60% | 40.87 | 55.13 | 23.02% |
| TrendGCN | 18.81 | 30.68 | 12.25% | 23.83 | 37.10 | 17.53% | 27.35 | 43.10 | 19.32% | 27.90 | 44.83 | 20.38% |
| STExplainer-CGIB | 19.14 | 30.77 | 12.91% | 25.76 | 38.36 | 16.05% | 31.72 | 48.51 | 17.98% | 28.43 | 44.69 | **16.85%** |
| STExplainer | 18.57 | **30.14** | **12.13%** | 24.48 | 36.78 | 15.89% | 31.39 | 47.18 | 18.05% | 29.60 | 46.41 | 17.37% |
| STGKD | 18.69 | 30.46 | 12.34% | 24.35 | 37.06 | 16.31% | 28.53 | 44.74 | 17.66% | 29.24 | 46.28 | 18.63% |
| BiTGraph | 18.82 | 30.44 | 12.25% | 24.73 | 37.08 | 16.03% | 31.65 | 47.52 | 18.20% | 29.85 | 46.75 | 17.50% |
| STC-Dropout | 18.75 | 30.38 | 12.33% | 26.85 | 39.32 | 16.50% | 34.15 | 51.22 | 20.54% | 33.74 | 50.37 | 19.98% |
| STG-NCDE | 19.21 | 31.09 | 12.76% | 24.82 | 37.41 | 17.30% | 29.24 | 44.17 | 19.44% | 30.97 | 47.19 | 20.86% |
| FreTS | 18.77 | 30.45 | 12.25% | 24.68 | 37.05 | 16.00% | 31.60 | 47.45 | 18.17% | 29.82 | 46.65 | 17.48% |
| **RSTIB-MLP** | **18.46** | **30.14** | 12.22% | **23.64** | **36.44** | **15.22%** | **27.15** | **42.85** | **17.19%** | **27.16** | **43.43** | 17.76% |
| Dataset | PEMS07 | | | | | | | | | | | |
| STID | 20.41 | 33.68 | 8.74% | 27.99 | 45.02 | 12.37% | 31.83 | 55.26 | 13.62% | 32.38 | 57.29 | 14.07% |
| GWN | 20.25 | **33.32** | 8.63% | 28.25 | 45.47 | 12.51% | 32.15 | 55.81 | 13.76% | 37.71 | 59.86 | 25.21% |
| TrendGCN | 20.43 | 34.32 | 8.51% | 26.87 | 44.65 | 14.59% | 31.94 | 55.28 | 20.78% | 36.78 | 57.89 | 23.22% |
| STExplainer-CGIB | 20.55 | 35.12 | 8.61% | 28.14 | 44.07 | 12.18% | 34.92 | 57.60 | 14.22% | 35.12 | 59.17 | 16.78% |
| STExplainer | 20.00 | 33.45 | 8.50% | 28.30 | 44.21 | 12.22% | 31.58 | **54.03** | 14.82% | 32.52 | 57.64 | 15.48% |
| STGKD | 20.30 | 34.30 | 8.87% | 27.04 | 43.83 | 12.23% | 31.64 | 55.16 | 13.69% | 32.16 | 56.89 | 14.08% |
| BiTGraph | 20.25 | 33.75 | 8.60% | 28.55 | 44.55 | 12.35% | 31.84 | 54.37 | 14.94% | 32.77 | 58.00 | 15.60% |
| STC-Dropout | 20.47 | 33.91 | 8.75% | 28.63 | 44.67 | 12.41% | 31.72 | 54.42 | 15.01% | 32.91 | 58.13 | 15.69% |
| STG-NCDE | 20.53 | 33.84 | 8.80% | 28.79 | 44.62 | 14.22% | 32.21 | 56.23 | 15.78% | 33.48 | 58.83 | 16.78% |
| FreTS | 19.92 | 33.65 | 8.70% | 28.60 | 44.40 | 14.10% | 32.05 | 56.00 | 15.65% | 33.30 | 58.60 | 16.65% |
| **RSTIB-MLP** | **19.84** | 33.90 | **8.33%** | **26.55** | **43.77** | **11.37%** | **30.15** | 54.08 | **12.65%** | **30.94** | **56.79** | **12.91%** |
| Dataset | PEMS08 | | | | | | | | | | | |
| STID | 14.87 | 23.97 | 10.43% | 20.26 | 32.24 | 14.05% | 26.64 | 45.73 | 15.63% | 27.76 | 48.64 | 16.45% |
| GWN | 14.67 | **23.49** | 9.52% | 20.52 | 32.66 | 14.19% | 26.91 | 46.19 | 15.78% | 28.04 | 49.13 | 16.61% |
| TrendGCN | 15.15 | 24.26 | 9.51% | 20.81 | 32.49 | 14.92% | 24.74 | 41.46 | 23.74% | 26.90 | 45.69 | 22.95% |
| STExplainer-CGIB | 14.87 | 24.07 | 10.26% | 23.66 | 35.49 | 24.34% | 24.87 | 43.14 | 15.32% | 26.50 | 44.62 | 15.54% |
| STExplainer | 14.59 | 23.91 | 9.80% | 20.28 | 32.66 | 13.37% | 25.42 | 43.41 | 16.77% | 27.17 | 45.79 | 15.26% |
| STGKD | 15.13 | 24.80 | 10.66% | 20.62 | 32.45 | 14.99% | 25.63 | 43.29 | 16.03% | 25.93 | 44.03 | 16.59% |
| BiTGraph | 14.85 | 24.20 | 9.90% | 20.55 | 33.15 | 13.50% | 25.70 | 43.75 | 16.90% | 27.45 | 46.10 | 15.40% |
| STC-Dropout | 14.70 | 24.32 | 9.75% | 20.35 | 32.82 | 13.95% | 25.55 | 44.02 | 15.25% | 26.75 | 47.15 | 16.15% |
| STG-NCDE | 15.45 | 24.81 | 9.92% | 21.36 | 33.25 | 15.23% | 28.35 | 41.89 | 16.33% | 29.44 | 47.32 | 18.62% |
| FreTS | 14.85 | 24.15 | 9.89% | 20.52 | 33.12 | 13.45% | 25.68 | 43.75 | 16.88% | 27.41 | 46.05 | 15.35% |
| **RSTIB-MLP** | **14.51** | 24.18 | **9.44%** | **19.90** | **31.86** | **12.92%** | **23.16** | **40.46** | 14.26% | **24.37** | 43.77 | 14.36% |
| Dataset | LargeST(SD) | | | | | | | | | | | |
| STID | 17.60 | 29.05 | 11.92% | 26.53 | 40.35 | 16.91% | 34.82 | 54.03 | 20.62% | 35.21 | 55.26 | 21.52% |
| GWN | 17.74 | 29.62 | 11.88% | 27.39 | 40.95 | 17.81% | 32.87 | 55.64 | 18.84% | 37.32 | 58.25 | 23.23% |
| TrendGCN | 17.39 | 29.63 | 11.64% | 25.84 | 39.64 | 16.23% | 31.45 | 51.71 | 17.83% | 33.63 | 52.18 | 18.85% |
| STExplainer-CGIB | 18.60 | 30.29 | 12.69% | 26.17 | 40.46 | 17.55% | 32.11 | 53.39 | 18.37% | 34.56 | 52.88 | 19.43% |
| STExplainer | 17.51 | 28.86 | 11.57% | 25.68 | 39.48 | 16.24% | 31.41 | 51.49 | 17.87% | 33.39 | 51.96 | 18.80% |
| STGKD | 17.60 | 29.42 | 11.62% | 25.85 | 39.71 | 16.08% | 31.52 | 51.37 | 17.67% | 33.93 | 52.67 | 18.97% |
| BiTGraph | 18.85 | 29.80 | 12.68% | 25.81 | 39.34 | 16.23% | 31.16 | 52.13 | 17.74% | 33.74 | 51.98 | 18.86% |
| STC-Dropout | 17.55 | 29.36 | 11.68% | 25.78 | 39.64 | 16.14% | 31.48 | 51.42 | 17.73% | 33.87 | 52.12 | 18.92% |
| STG-NCDE | 17.58 | 29.14 | 11.87% | 26.24 | 40.39 | 16.52% | 31.83 | 52.67 | 17.86% | 33.76 | 52.23 | 18.97% |
| FreTS | 17.54 | 29.01 | 11.95% | 26.16 | 40.21 | 16.53% | 31.60 | 52.99 | 17.71% | 34.09 | 52.51 | 18.86% |
| **RSTIB-MLP** | **17.50** | **28.75** | **11.20%** | **25.02** | **38.37** | **15.42%** | **30.60** | **50.52** | **16.85%** | **32.78** | **50.38** | **17.92%** |
| Dataset | Weather2K-R | | | | | | | | | | | |
| STID | 3997.92 | 6199.77 | 65.34% | 4950.47 | 6610.89 | 67.06% | 6301.76 | 8071.43 | 76.94% | 7654.47 | 9660.18 | 82.08% |
| GWN | 3991.24 | 6207.50 | 66.00% | 5218.42 | 6896.87 | 66.72% | 6883.07 | 8681.24 | 74.27% | 8324.23 | 9832.49 | 83.08% |
| TrendGCN | 3987.92 | 6223.53 | 65.30% | 4589.04 | 6274.23 | 63.31% | 5982.71 | 7688.16 | 72.99% | 7108.36 | 8964.54 | 81.53% |
| STExplainer-CGIB | 3994.82 | 6200.83 | 65.35% | 4789.03 | 6540.73 | 67.32% | 6215.86 | 7985.24 | 75.72% | 7775.50 | 9812.44 | 81.61% |
| STExplainer | 3992.57 | 6198.33 | 65.22% | 4786.53 | 6537.73 | 67.18% | 6213.36 | 7982.24 | 75.58% | 7773.00 | 9809.44 | 81.47% |
| STGKD | 3990.07 | 6195.83 | 65.08% | 4784.03 | 6534.73 | 67.02% | 6210.86 | 7979.24 | 75.44% | 7770.50 | 9805.94 | 81.33% |
| BiTGraph | 3989.32 | 6216.03 | 65.21% | 4588.67 | 6274.45 | 63.15% | 5981.76 | 7680.34 | 72.91% | 7103.92 | 8964.87 | 81.38% |
| STC-Dropout | 3986.43 | 6205.21 | 65.35% | 4792.87 | 6543.12 | 67.25% | 6205.49 | 7975.08 | 75.45% | 7782.56 | 9817.33 | 81.60% |
| STG-NCDE | 3992.57 | 6199.03 | 65.22% | 4787.33 | 6538.53 | 67.15% | 6214.36 | 7983.04 | 75.58% | 7774.00 | 9810.24 | 81.46% |
| FreTS | 3984.37 | 6219.03 | 65.12% | 4585.09 | 6269.53 | 63.13% | 5978.26 | 7683.56 | 72.81% | 7104.06 | 8959.94 | 81.35% |
| **RSTIB-MLP** | **3964.53** | **6191.08** | **64.94%** | **4561.97** | **6239.33** | **62.96%** | **5948.52** | **7645.33** | **72.64%** | **7073.69** | **8914.37** | **81.17%** |
| Dataset | Electricity | | | | | | | | | | | |
| STID | 20.18 | 39.82 | 15.92% | 26.08 | 47.98 | 21.74% | 37.25 | 65.27 | 28.23% | 50.97 | 81.16 | 45.78% |
| GWN | - | - | - | - | - | - | - | - | - | - | - | - |
| TrendGCN | 19.98 | **39.62** | **15.72%** | 25.23 | 46.48 | 20.37% | 34.35 | 63.82 | 26.78% | 47.26 | 78.65 | 42.38% |
| STExplainer-CGIB | - | - | - | - | - | - | - | - | - | - | - | - |
| STExplainer | - | - | - | - | - | - | - | - | - | - | - | - |
| STGKD | 20.15 | 40.05 | 15.89% | 25.40 | 46.75 | 20.65% | 34.90 | 64.90 | 27.65% | 48.75 | 79.80 | 44.90% |
| BiTGraph | 19.98 | 39.87 | 16.12% | 25.52 | 47.10 | 21.05% | 35.68 | 65.82 | 28.55% | 49.78 | 78.95 | 43.78% |
| STC-Dropout | 19.92 | 39.85 | 16.47% | 26.12 | 48.25 | 21.69% | 36.68 | 68.32 | 30.27% | 49.30 | 78.03 | 42.78% |
| STG-NCDE | 19.85 | 39.92 | 16.52% | 26.05 | 48.12 | 21.78% | 36.75 | 68.45 | 30.18% | 49.23 | 77.90 | 42.85% |
| FreTS | 20.12 | 40.45 | 16.22% | 26.15 | 47.88 | 21.95% | 37.30 | 69.25 | 31.05% | 52.98 | 78.75 | 43.78% |
| **RSTIB-MLP** | **19.80** | 39.67 | **15.72%** | **24.50** | **45.85** | **19.95%** | **33.80** | **62.50** | **25.85%** | **45.30** | **74.60** | **40.75%** |

**Robustness Study.** We evaluate the robustness of RSTIB-MLP by injecting noise into both the input and the target area, similar to the empirical study we conduct for evaluating the harmful aspect of dual noise effect. As presented in Table 2, the results demonstrate that the increase in errors for RSTIB-MLP is significantly lower compared to existing robust methods when dealing with noisy data. This finding underscores the robustness of RSTIB-MLP, which can be attributed to its consideration of both noisy input and target information conveyed. In contrast, previous approaches typically consider only a single noisy area. Our method's ability to handle both noisy patterns contributes to its enhanced robustness.

**Learning with Clean Data.** In this analysis, we investigate the behavior of RSTIB-MLP when learning with clean data. As depicted in Table 2 and Table 11 in Appendix K.1, we observe some improvements in performance metrics on clean datasets, although not significant. It is noteworthy that while our primary objective is to enhance robustness on noisy datasets, the observed slight improvement on clean datasets suggests potential benefits. However, we hypothesize that the marginal improvement could stem from the challenge of effectively balancing the informative terms within the learning objective, thus impacting the overall performance.

**Inspecting Representation Learning from a Feature Variance Perspective: A Case Study.** In this case study, we examine the superiority of our method from the perspective of feature variance, a crucial aspect for effective model evaluation. As discussed in (Bardes et al., 2021), maintaining feature diversity is essential to mitigate feature collapse and enhance model robustness. The quantitative case findings in Fig.3 indicate that our proposed knowledge distillation module significantly boosts feature variance, a critical factor in capturing the intricate and dynamic spatial-temporal patterns. This observation underscores the effectiveness of accounting for the noise impact on different time series when balancing the infor-

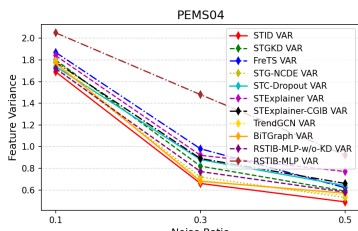

**Figure 3: Feature Variance (Var) of Different Methods w.r.t different noise ratio ($\gamma$) in PEMS04 Dataset**

mative terms in the learning objective, which is achieved by incorporating knowledge distillation into the training regime. We also provide a model interpretation case study to visualize the distribution of learned representation in *Appendix. K.8.*

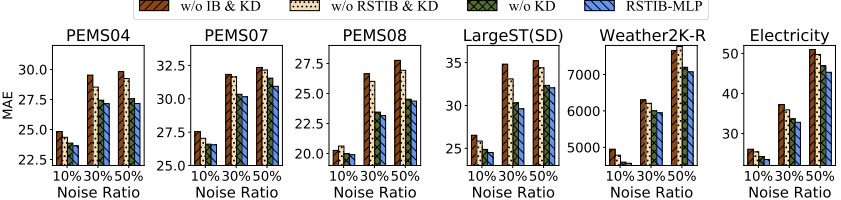

**Figure 4: Ablation Study Results on Different Benchmark Datasets When Combating Noises with Different Noise Ratios**

## 5.2 ABLATION STUDY

We assess our proposed components within RSTIB-MLP through its various variants: i) "**w/o IB & KD**": Excludes any Information Bottleneck (IB)-based enhancement and the knowledge distillation module. ii) "**w/o RSTIB & KD**": Similar to the first variant but implements vanilla IB. iii) "**w/o KD**": Removes the knowledge distillation module while instantiating the RSTIB principle. The ablation study, as shown in Fig.4, reveals significant performance degradation without IB instantiation in most scenarios, emphasizing its role in mitigating the detrimental effects of spatial-temporal data noise. However, the results also show some circumstances where implementing the vanilla IB principle results in even worse performance. The potential reason is that vanilla IB has not considered noisy information conveyed by the target, while also challenging to balance the informative terms within its objective. Besides, instantiating RSTIB can make significant performance improvements compared with vanilla IB instantiation or non-IB enhanced instantiation. This underscores the importance of minimizing the noisy information conveyed by both input and target data ends in the spatio-temporal

forecasting scenario. Furthermore, the knowledge distillation module contributes to performance enhancement, due to the better balance of the informative terms during robust representation learning.

### 5.3 HYPERPARAMETER ANALYSIS

We conduct a hyperparameter analysis focusing on two key hyperparameters in RSTIB-MLP: the distance function for computing the noise impact indicator $\hat{\alpha}$ and the Lagrange multipliers $\lambda$ ($\lambda_x, \lambda_y, \lambda_z$). This study aims to evaluate their influence on model performance using the PEMS04 dataset (results shown in Fig. 5): i) **Distance Function for Noise Impact**: We test Mean Absolute Error (MAE), Mean Squared Error (MSE), and Smooth L1 Loss for computing $\hat{\alpha}$. The MAE distance function provides the best

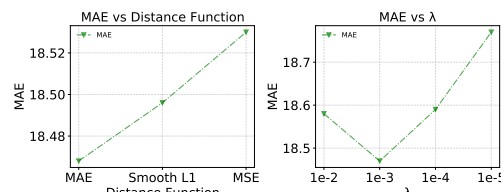

**Figure 5: Hyperparameter analysis showing comparisons for distance function (left) and $\lambda$ (right)**

MAE metric. ii) **The Lagrange Multipliers**: We set $\lambda_x, \lambda_y, \lambda_z$ to equal values, varying them over $1 \times 10^{-2}, 1 \times 10^{-3}, 1 \times 10^{-4}, 1 \times 10^{-5}$. The best MAE value is achieved at $\lambda = 1 \times 10^{-3}$. The detailed analysis regarding MAE, RMSE, MAPE metrics have been included on Appendix K.9.

### 5.4 COMPUTATIONAL EFFICIENCY

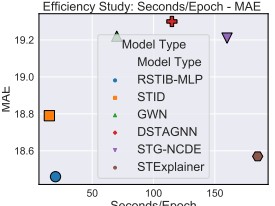

**Figure 6: Efficiency Study in PEMS04 Dataset**

This section compares the efficiency of RSTIB-MLP with some representative state-of-the-art STGNN-based methods. We also include an MLP-based method, STID (Shao et al., 2022a), as an MLP-based baseline. We measure the efficiency by recording the average training time per epoch of all methods on the PEMS04 dataset. All evaluations are conducted on an NVIDIA RTX 3090Ti GPU. Fig.6 displays the results. We can see that prior STGNN-based works require more time due to the sophisticated model design (See Section H). By contrast, our work utilizes computationally more efficient MLP networks, resulting in a more streamlined model architecture, allowing faster processing and shorter training time.

Beyond the results provided in Fig.6, we also evaluate the overall training time to convergence for better showcasing our method's superior efficiency. We include the results in Table 3, in which it is clear that our full training is much faster than the competing methods. For example, as shown in Table 3, our method reduces up to 88.42% of training convergence time compared to one of the most effective STGNN-based baseline methods, STExplainer (Tang et al., 2023).

**Table 3: Training convergence time for different baselines.**

| Method | Total Training Time (Seconds) |
|---|---|
| RSTIB-MLP | 2842.3 |
| DSTAGNN | 9283.7 |
| Graph-WaveNet | 7308.6 |
| STG-NCDE | 9238.7 |
| STExplainer | 24514.0 |

## 6 CONCLUSION

In noisy spatial-temporal forecasting scenarios, noise perturbation can degrade forecasting accuracy and induce feature collapse. We propose the *Robust Spatial-Temporal Information Bottleneck (RSTIB)* principle for guiding robust representation learning to mitigate these effects. By leveraging RSTIB, we instantiate our method using a pure MLP network, resulting in a computationally efficient and robust RSTIB-MLP model for the task. Additionally, we incorporate a knowledge distillation module into our training regime. Knowledge distillation can enhance feature diversity and improve predictive accuracy by better leveraging the knowledge from previously trained teacher models to balance informative terms within the objective of RSTIB-MLP. Through comprehensive evaluation encompassing feature variance and predictive performance metrics, our approach demonstrates superior performance in handling of noise. It maintains robust forecasting accuracy under challenging conditions while computationally more efficient than stat-of-the-art STGNN-based methods.

## REPRODUCIBILITY STATEMENT

For enhancing reproducibility, we provide the links to all the datasets in *Appendix.* E. For the theoretical results, detailed proofs has been provided in *Appendix.*F, along with the assumptions made in *Assumption.* 4.1 and *Assumption.* 4.2. Implementation details for RSTIB-MLP are also provided in *Appendix.* J.1. Besides, settings for each dataset are detailed in the **Datasets** and **Implementation** subsections in Section 5.

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

# Appendix

## A  NOTATIONS

Our notations are elaborated in Table 4.

## B  ADDITIONAL PRELIMINARIES

### B.1  MATHEMATICAL PRELIMINARIES AND DEFINITIONS

This section provides mathematical preliminaries concerning entropy and mutual information using three discrete random variables $X$, $Y$, and $Z$ for illustrative purposes. It is important to note that these variables do not carry specific meanings within this context and the notations used here are distinct from those in the main discussion. Additionally, this section offers an intuitive understanding of each term.

**Definition B.1.** *Entropy. We define the entropy $H(X)$ of a discrete random variable $X$ as a measure of its uncertainty, using its marginal distribution $p(x)$. Mathematically, entropy is expressed as:*

$$H(X) = -\sum_{x \in X} p(x) \log p(x), \tag{7}$$

*where the summation extends over all possible outcomes $x$ of the random variable $X$. The function $H(X)$ quantifies the expected information content or uncertainty inherent in $X$'s outcomes.*

**Definition B.2.** *Joint Entropy. The entropy of two random variables $X$ and $Y$ can be jointly considered by viewing them as components of a single vector-valued random variable. This joint entropy is defined as:*

$$H(X, Y) = -\sum_{x \in X, y \in Y} p(x, y) \log p(x, y)$$
$$= -\sum_{x \in X} \sum_{y \in Y} p(x, y) \log p(x, y), \tag{8}$$

*where $p(x, y)$ represents the joint probability distribution of $X$ and $Y$. This definition encapsulates the total uncertainty present when considering the distribution of both variables simultaneously.*

**Definition B.3.** *Conditional Entropy. Given two discrete random variables $X$ and $Y$, the conditional entropy of $X$ given $Y$ is defined as:*

$$H(X|Y) = -\sum_{y \in Y} p(y) \sum_{x \in X} p(x|y) \log p(x|y), \tag{9}$$

*where $p(x|y)$ is the conditional probability of $X$ given $Y$, and $p(y)$ is the marginal distribution of $Y$. A value of $H(X|Y) = 0$ implies that knowing $Y$ completely determines $X$, signifying no remaining uncertainty about $X$ once $Y$ is observed.*

*This concept allows us to understand $H(X)$ as a priori entropy of $X$, while $H(X|Y)$ represents a posteriori entropy—reflecting the uncertainty in $X$ after $Y$ is known. The reduction in entropy, $H(X) - H(X|Y)$, quantifies the amount of information $Y$ provides about $X$, which is formally termed mutual information in **Definition B.4**.*

**Table 4: Notations**

| Symbol | Description |
|---|---|
| $N$ | The number of time series($i.e.$, nodes). |
| $P$ | The length of historical input. |
| $F$ | The length of forecasting target. |
| $C$ | The number of features in each input or target time series at a specific time slot. |
| $X^h$ | The historical spatial-temporal data $X^h \in \mathbb{R}^{P \times N \times C}$, with $N$ time series of $C$ features in each time series within $P$ nearest historical time slots, with each $X^h_{t,i} \in \mathbb{R}^C$. |
| $X$ | The input to RSTIB-MLP model, generated from $X^h$. The dimension of $X$, along with $X_{t,i}$, depends on $X_h$ and the attachment of spatial-temporal information from a specially designed module. |
| $Y$ | The forecasting target data $Y \in \mathbb{R}^{F \times N \times C}$ with N time series of C features in each time series within $F$ nearest future time slots, with each $Y_{t,i} \in \mathbb{R}^C$. |
| $\tilde{X}$ | The reparameterized input. |
| $\tilde{Y}$ | The reparameterized target. |
| $Y^T$ | The teacher model's output. |
| $Y^S$ | The RSTIB-MLP model's output. |
| $d$ | The hidden dimension of each $z \in Z$, $i.e.$, $z_i \in \mathbb{R}^d$ for $i = 1, \ldots, N$. |
| $Z$ | The encoded spatial-temporal representation, comprised of a series of latent spatial-temporal representations $(z_1, z_2, \ldots, z_N)$ where $z_i \in \mathbb{R}^d$ for $i = 1, \ldots, N$. $Z \in \mathbb{R}^{N \times d}$. |
| **Cov** | The covariance matrix of $Z$, with each $\mathbf{Cov}_{ii}$ representing the variance of the i-th feature across the representations, $i.e.$, the diagonal elements of **Cov**. |
| **Var** | The feature variance defined in Eq.(1). |
| $D$ | The distance function for calculating the noise impact indicators. |
| $\hat{\alpha}$ | The noise impact indicator, where $\hat{\alpha}_i$ is computed for each time series. |
| $\beta$ | The Lagrange multiplier defining the trade-off between the compression of $X$ and preservation of $Y$ in the IB objective. |
| $\beta_1, \beta_2$ | The Lagrange multipliers defining the informative terms within the RSTIB objective. |
| $\lambda_x, \lambda_y, \lambda_z$ | The Lagrange multipliers defining the balance between the informative terms within the RSTIB-MLP objective. |
| $\mathcal{L}_{IB}$ | The original objective of IB principle. |
| $\mathcal{L}_{RSTIB}$ | The objective of RSTIB principle. |
| $\mathcal{L}_{RSTIB-MLP}$ | The learning objective of RSTIB-MLP. |
| $\mathcal{L}_{reg}$ | The typical regression loss. |
| $\mathcal{L}_x, \mathcal{L}_y, \mathcal{L}_z$ | The upper bounds of input regularization, target regularization, representation regularization. |
| $L$ | The number of layers. |
| $N_d$ | The time slots in a day. |
| $N_w$ | The number of days in a week. |
| $f_T$ | The teacher model. |
| $\eta$ | The learning rate. |
| $\mathbb{E}$ | The expectation of a random variable, $i.e.$, the mean of the possible values a random variable can take, weighted by the probability of those outcomes. |
| $E$ | The maximum epoch number. |
| $B$ | The batch size. |
| $r$ | The decay factor. |
| $\tau$ | The non-linear activation. |
| $\gamma$ | The noise ratio. |
| $H(\cdot)$ | The entropy of a discrete random variable, $e.g.$, $H(X)$ represents the entropy of $X$. |
| $I(\cdot, \cdot)$ | The mutual information between two discrete random variables, $e.g.$, $I(X; Y)$ represents the mutual information between $X$ and $Y$. |
| $LI(\cdot, \cdot)$ | The lautum information between two discrete random variables, $e.g.$, $LI(X; Y)$ represents the lautum information between $X$ and $Y$. |

**Remark B.1.** *Conditional Entropy w.r.t three variables. The conditional entropy $H(Z|X,Y)$ quantifies the residual uncertainty in a random variable $Z$ when the values of other variables $X$ and $Y$ are known. It is mathematically defined as:*

$$H(Z|X,Y) = -\sum_{x \in X, y \in Y} p(x,y) \sum_{z \in Z} p(z|x,y) \log p(z|x,y) \tag{10}$$

*Here, $p(x,y)$ represents the joint distribution of $X$ and $Y$, and $p(z|x,y)$ is the conditional probability of $Z$ given that $X$ and $Y$ take the values $x$ and $y$ respectively. This measure effectively describes how much uncertainty in $Z$ remains after observing both $X$ and $Y$.*

**Definition B.4.** *Mutual Information. Given two discrete random variables $X$ and $Y$, their mutual information (MI), denoted as $I(X;Y)$, is defined by:*

$$
\begin{aligned}
I(X;Y) &= H(X) - H(X|Y) \\
&= -\sum_{x \in X} p(x) \log p(x) + \sum_{x \in X, y \in Y} p(x,y) \log p(x|y) \\
&= -\sum_{x \in X} \log p(x) \sum_{y \in Y} p(x,y) + \sum_{x \in X, y \in Y} p(x,y) \log p(x|y) \\
&= \sum_{x \in X, y \in Y} p(x,y) \log \left( \frac{p(x|y)}{p(x)} \right),
\end{aligned}
\tag{11}
$$

*where $p(x,y)$ is the joint distribution between $X$ and $Y$, $p(x)$ is the marginal distribution of $X$ and $p(x|y)$ is the conditional probability distribution of $X$ given $Y$, respectively.*

**Definition B.5.** *Relative Entropy. The relative entropy, or Kullback-Leibler (KL) distance, between two probability mass functions $p(x)$ and $q(x)$ is defined as follows:*

$$KL(p||q) = \sum_{x \in X} p(x) \log \frac{p(x)}{q(x)}. \tag{12}$$

*The mutual information between $X$ and $Y$ can also be expressed as $I(X;Y) = KL(p(x,y)||p(x)p(y))$, which implies that mutual information is the relative entropy between the joint distribution $p(x,y)$ and the product of the marginal distributions $p(x)p(y)$.*

**Remark B.2.** *Mutual information satisfies the following identities:*

$$I(X;Y) = H(X) - H(X|Y) = H(Y) - H(Y|X) = I(Y;X) \tag{13}$$

$$I(X;Y) = H(X) - H(X|Y) = H(X) + H(Y) - H(X,Y). \tag{14}$$

*The relationships among $H(X)$, $H(Y)$, $H(X|Y)$, $H(Y|X)$, $I(X;Y)$, and $H(X,Y)$ can be visualized in a Venn diagram, as shown in Fig. 7.*

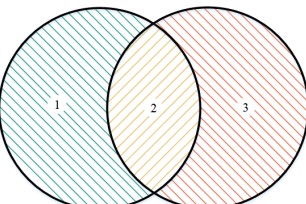

**Figure 7: Relationship between** $H(X)$**,** $H(Y)$**,** $H(X|Y)$**,** $H(Y|X)$**,** $I(X;Y)$**,** $H(X,Y)$**. (1):** $H(X|Y)$**; (2):** $I(X;Y)$**; (3):**$H(Y|X)$**; (1+2):** $H(X)$**; (2+3):** $H(Y)$**; (1+2+3):** $H(X,Y)$**.**

**Remark B.3.** *Mutual Information w.r.t to three variables $I(Z;X,Y)$. The mutual information $I(Z;X,Y)$ quantifies the shared information between the variable $Z$ and the variables consisted of both $X$ and $Y$. It is defined mathematically as:*

$$I(Z;X,Y) = \sum_{x \in X, y \in Y, z \in Z} p(x,y,z) \log \frac{p(z|x,y)}{p(z)} \tag{15}$$

*where $p(x,y,z)$ represents the joint probability distribution of the variables $X$, $Y$, and $Z$. This expression highlights how much uncertainty in $Z$ is reduced by knowing both $X$ and $Y$.*

**Definition B.6.** *Lautum Information. Because of the non-symmetry of the KL divergence, Lautum information is defined as the divergence from the product of the marginal distributions to the joint distribution of two random variables $X$ and $Y$ and is given by:*

$$LI(X;Y) = KL(p(x)p(y) \parallel p(x,y)), \tag{16}$$

*where $p(x,y)$ is the joint distribution of $X$ and $Y$. $p(x)$ and $p(y)$ are the marginal distributions of $X$ and $Y$ respectively. This concept was introduced by (Palomar & Verdú, 2008).*

**Definition B.7.** *Conditional Mutual Information. The conditional mutual information between $X$ and $Y$ given $Z$, denoted as $I(X;Y|Z)$, measures the amount of information shared between $X$ and $Y$ that is unique and not already explained by $Z$. It is defined as:*

$$
\begin{aligned}
I(X;Y|Z) &= H(X|Z) - H(X|Y,Z) \\
&= \sum_{x \in X, y \in Y, z \in Z} p(x,y,z) \log\left(\frac{p(x,y|z)}{p(x|z)p(y|z)}\right) \\
&= \sum_{x \in X, y \in Y, z \in Z} p(x,y,z) \log\left(\frac{p(x|y,z)}{p(x|z)}\right).
\end{aligned}
\tag{17}
$$

*which quantifies the additional information about $X$ obtained by observing $Y$ when the influence of $Z$ is already known.*

**Definition B.8.** *Interaction Information. The interaction information concerning the variables $X$, $Y$, and $Z$ quantifies the unique information shared by these three variables. It is formally defined as:*

$$I(X;Y;Z) = I(X;Y) - I(X;Y|Z) \tag{18}$$

*This measure reveals whether the mutual information between $X$ and $Y$ is increased or decreased by conditioning on $Z$.*

### B.2 PRELIMINARIES FOR IB AND DVIB

Here, we detail the preliminaries regarding the Information Bottleneck(IB) and Deep Variational Information Bottleneck(DVIB). We denote X as the input to different IB models, Z as the encoding from X, and Y as the target.

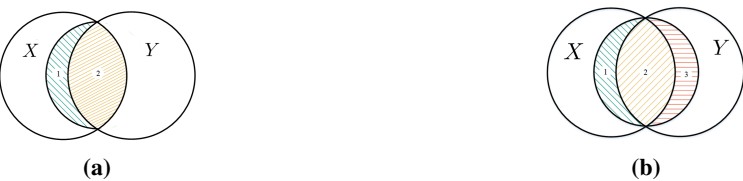

**(a)**          **(b)**

**Figure 8: Comparison of IB(a) and DVIB with lifted markov assumption $Z - X - Y$(b). Refer to Fig. 2 for more details.**

In Fig. 8a, the entropy of $X$, *i.e.*, $H(X)$, and the entropy of $Y$, *i.e.*, $H(Y)$, are depicted as circles, with their mutual information $I(X;Y)$ represented in the overlapping area. The representation learning guided by the IB principle aims to optimize the information flow by retaining as much relevant information about $Y$ in $Z$ as possible while minimizing the redundant information from $X$. This principle targets reducing the irrelevant information $H(X|Y)$ $Z$ captures, namely $I(X;Z|Y)$, aiming for what is termed the "*minimal sufficient representation*", ideally encapsulating solely $I(X;Y)$. Achieving this optimal representation presents substantial challenges due to the intrinsic complexities of the models and the varied selection of parameters and hyperparameters, such as $\beta$ in Eq. (2).

Incorporating the IB model with deep learning, where mutual information terms are modeled using deep neural networks (DNNs), has proven successful. The DVIB method leverages deep learning to approximate the IB model, finding a sufficient statistic $Z$ given $X$ while retaining pertinent side information about $Y$. The approach involves parameterizing the conditional probabilities $P(Z|X)$ and $P(Y|Z)$ using DNNs, thus enabling direct recovery of the terms in the original IB objectives.

Regarding the assumptions of the Markov chain, the typical practice in the original IB formulation assumes $Z - X - Y$. This assumption is also utilized to derive DVIB. Additionally, by its construction, the DVIB model satisfies the data generating process, which implies that the Markov assumption $X - Z - Y$ holds. Adhering to both Markov chain restrictions in DVIB may seem overly restrictive, and as pointed out by (Wieczorek & Roth, 2020), no directed acyclic graph (DAG) with three vertices can faithfully represent such a distribution. Consequently, Wieczorek & Roth (2020) theoretically explore the possibility of relaxing the $Z - X - Y$ restriction by demonstrating how $I(Z; Y)$ can be lower bounded, thus potentially circumventing the necessity for the $Z - X - Y$ configuration. As illustrated in Fig. 8a, the original IB method does not encompass a region representing $I(Z; Y|X)$, owing to its reliance on the $Z - X - Y$ Markov chain assumption. Conversely, in the DVIB approach with this assumption lifted, the term $I(Z; Y|X) \neq 0$ is represented in Fig.8b. Detailed explanations regarding the derivation are provided in *Proofs. F.5*.

---

**Algorithm 1** RSTIB-MLP for Spatial-Temporal Forecasting

ht!

**Input:** Historical spatial-temporal data $X^h$, input adjacency matrix $A$(optional), trained teacher model $f_T$, $N$ time series, $N_d$ time slots in a day, $N_w = 7$ days in a week, Lagrange multipliers $\lambda_x, \lambda_y, \lambda_z$, maximum epoch number $E$, learning rate $\eta$.

1: **for** $e = 1$ to $E$ **do**
2:     *// Obtain noise impact indicator for each time series*
3:     Obtain teacher output $Y^T = f_T(A, X^h)$.
4:     Calculate noise impact indicator $\hat{\alpha}_i$ for each time series $i$ according to Eq. (5).
5:     *// Prepare the input $X$ to the RSTIB-MLP*
6:     Attach the spatial-temporal information to the historical input data $X_h$, according to Eq. (20), to obtain the input $X$ to RSTIB-MLP.
7:     *// Data Reparameterization for obtaining $\tilde{X}$ and $\tilde{Y}$*
8:     Adopt the MLP encoders, along with two additional Fully-Connected(FC) layers to align with the dimension of $X$ and $Y$, respectively. They are utilized for parameterizing $P_{\hat{\phi}_x} \sim \mathcal{N}(\hat{\mu_x}, \hat{\sigma_x}^2)$ and $P_{\hat{\phi}_y} \sim \mathcal{N}(\hat{\mu_y}, \hat{\sigma_y}^2)$.
9:     Establish $P_{\phi_x} \sim \mathcal{N}(\mu_x, \sigma_x^2)$ and $P_{\phi_y} \sim \mathcal{N}(\mu_y, \sigma_y^2)$, where $\mu_x = x + \hat{\mu_x}$, $\mu_y = y + \hat{\mu_y}$, $\sigma_x^2 = \hat{\sigma_x}^2$ and $\sigma_y^2 = \hat{\sigma_y}^2$ respectively.
10:     Adopt data reparameterization to obtain $\tilde{X}$ and $\tilde{Y}$, by obtaining $\tilde{x} = \mu_x + \sigma_x \epsilon$ and $\tilde{y} = \mu_y + \sigma_y \epsilon$, with each $\tilde{x} \in \tilde{X}$ and $\tilde{y} \in \tilde{Y}$ respectively, and $\epsilon \sim \mathcal{N}(0, 1)$.
11:     *// Input Regularization and Target Regularization*
12:     Calculate the upper bounds of the input and target regularization according to **Proposition 4.3**.
13:     *// Data Reparameterization for obtaining $Z$*
14:     Adopt the same MLP encoders, sharing the same parameters, to parameterize the posterior distribution $P_{\phi_z} \sim \mathcal{N}(\mu_z, \sigma_z^2)$.
15:     Obtain $Z$ by obtaining $z = \mu_z + \sigma_z \epsilon$ through reparameterization, where $z \in Z$.
16:     *// Representation Regularization*
17:     Calculate the upper bound of the representation regularization according to **Proposition 4.4**.
18:     *// Decoder*
19:     Use a simple regression layer to obtain the output $Y^S$ according to $Z$.
20:     Calculate the total loss $\mathcal{L}_{RSTIB-MLP}$ according to Eq. (6).
21:     Update each parameter $\Theta$ in $\Theta$ as $\Theta = \Theta - \eta \cdot \nabla_\Theta \mathcal{L}_{RSTIB-MLP}$.
22: **end for**
23: **return** $\Theta$

---

### B.3 PRELIMINARIES FOR SAMPLE INDISTINGUISHABILITY

A recent work (Deng et al., 2021) identifies that the essential element for the efficacy of STGNNs lies in the capability of GCN to mitigate the issue of spatial indistinguishability. Thus, in MLP for spatial-temporal forecasting, additional modules are needed to alleviate the sample indistinguishability bottleneck by attaching the spatial-temporal information. In this study, spatial-temporal prompts (Tang

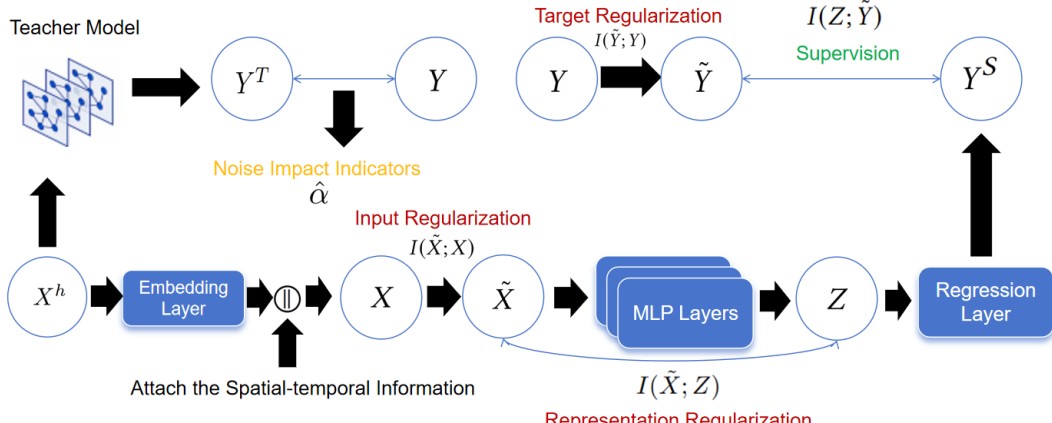

**Figure 9: The framework of RSTIB-MLP. The historical input data $X^h$ is attached with the spatial-temporal information to generate RSTIB-MLP's input $X$. Then, input regularization, target regularization and representation regularization are imposed, along with the optimization for supervision. $X^h$ is also used to calculate the noise impact indicators to quantify the noise impact on each time series to balance the informative terms within this framework better.**

et al., 2024), which is an extension of spatial-temporal identity (Shao et al., 2022a), are adopted to attach this information to the historical input data $X^h$ for obtaining the input $X$ to the models.

### B.3.1 SPATIAL-TEMPORAL IDENTITIES

With the spatial-temporal identities technique, inputs can be attached with spatial-temporal identity information, which is as follows:

$$X_{t,i} = \text{FC}(X_{t,i}^h)\|\mathbf{E}_i\|\mathbf{T}_t^{TiD}\|\mathbf{T}_t^{DiW}, \tag{19}$$

where FC refers to fully connected layers that map the dimension of the historical input data $X^h$ from $\mathbb{R}^{P\times N\times C}$ to the dimension $\mathbb{R}^{P\times N\times C'}$. Assuming $N$ time series, $N_d$ time slots in a day and $N_w = 7$ days in a week, the spatial-temporal identities are in three trainable embedding matrices, *i.e.*, $\mathbf{E} \in \mathbb{R}^{N\times C'}$ with each $\mathbf{E}_i \in \mathbb{R}^{C'}$, $\mathbf{T}^{TiD} \in \mathbb{R}^{N_d\times C'}$ with each $\mathbf{T}_t^{TiD} \in \mathbb{R}^{C'}$, and $\mathbf{T}^{DiW} \in \mathbb{R}^{N_w\times C'}$ with each $\mathbf{T}_t^{DiW} \in \mathbb{R}^{C'}$. The input to the model will be $X \in \mathbb{R}^{P\times N\times 4C'}$ by concatenating ($\|$) each term.

### B.3.2 SPATIAL-TEMPORAL PROMPTS

With the spatial-temporal prompts technique, inputs can be attached with spatial-temporal contextual information, including which is as follows:

$$X_{t,i} = \text{FC}_1(X_{t,i}^h)\|\text{FC}_2(\mathbf{E}_i^{(\alpha)})\|\text{FC}_3(\mathbf{E}_t^{(\beta)})\|\text{FC}_4(\mathbf{E}_t^{(ToD)})\|\text{FC}_5(\mathbf{E}_t^{(DoW)}). \tag{20}$$

Here, the terms $\mathbf{E}^{(\alpha)} \in \mathbb{R}^{N\times\hat{C}}$ with each $\mathbf{E}_i^{(\alpha)} \in \mathbb{R}^{\hat{C}}$ represents learnable spatial prompt, $\mathbf{E}^{(ToD)} \in \mathbb{R}^{N_d\times\hat{C}}$ with each $\mathbf{E}_t^{(ToD)} \in \mathbb{R}^{\hat{C}}$ and $\mathbf{E}^{(DoW)} \in \mathbb{R}^{N_w\times\hat{C}}$ with each $\mathbf{E}_t^{(DoW)} \in \mathbb{R}^{\hat{C}}$ represent the learnable temporal prompts, with the same settings that we have $N$ time series, $N_d$ time slots in a day and $N_w = 7$ days in a week. $\mathbf{E}_{t-P:t}^{(\beta)} \in \mathbb{R}^{P\times N\times\hat{C}}$ with each $\mathbf{E}_{t,i}^{(\beta)} \in \mathbb{R}^{\hat{C}}$ represents the dynamic spatio-temporal transitional prompt, inherent from (Han et al., 2021). $\text{FC}_i$, where $i = 1\ldots 5$, refers to fully connected layers that map the data and all the embeddings to the same dimension $C'$. In this case, the input fitted into the MLP networks will be $X \in \mathbb{R}^{P\times N\times 5C'}$, with each $X_{t,i} \in \mathbb{R}^{5C'}$.

## C ALGORITHM

Our learning framework is shown in Figure 9. Our algorithm is detailed in **Algorithm 1** .

## D  BASELINES

All baselines for comparisons are based on their original implementations. We list their source links here.

- STID, https://github.com/zezhishao/STID
- STExplainer, https://github.com/HKUDS/STExplainer
- TrendGCN, https://github.com/juyongjiang/TrendGCN
- STG-NCDE, https://github.com/jeongwhanchoi/STG-NCDE
- DSTAGNN, https://github.com/SYLan2019/DSTAGNN
- STGCN, https://github.com/VeritasYin/STGCN_IJCAI-18
- GWN, https://github.com/nnzhan/Graph-WaveNet
- AGCRN, https://github.com/LeiBAI/AGCRN
- GMSDR, https://github.com/dcliu99/MSDR
- FOGS, https://github.com/kevin-xuan/FOGS
- BiTGraph, https://github.com/chenxiaodanhit/BiTGraph
- FreTS, https://github.com/aikunyi/FreTS

## E  DATASETS

### Table 5: Statistics of Datasets

| Dataset | # Node | #Time Steps | #Sample Rate | #Time Span |
|---------|--------|-------------|--------------|------------|
| PEMS04 | 307 | 16992 | 5min | 01/2018 - 02/2018 |
| PEMS07 | 883 | 28224 | 5min | 05/2017 - 08/2017 |
| PEMS08 | 170 | 17856 | 5min | 07/2016 - 08/2016 |
| LargeST(SD) | 716 | 35040 | 15min | 01/2017 – 12/2021 |
| Weather2K-R | 1866 | 40896 | 1hour | 01/2017 – 08/2021 |
| Electricity | 336 | 2184 | 1hour | 10/2014 – 12/2014 |

The statistical information for six datasets is summarized in Table 5.

The **PEMS04/07/08** datasets are a comprehensive collection of traffic data gathered from Districts 4, 7, and 8 of Caltrans, respectively. These datasets typically include flow (vehicles per hour), speed (miles per hour), and occupancy (percentage of time the detector is occupied), recorded across multiple lanes and aggregated into 5-minute intervals. Public accessed data can be found in (Guo et al., 2021b): https://github.com/guoshnBJTU/ASTGNN/tree/main/data

The versions of the datasets are the same as the sources' default versions.

**LargeST** (Liu et al., 2024a): It is publicly available at https://github.com/liuxu77/LargeST.

**Weather2K-R** (Zhu et al., 2023b): It is publicly available at https://github.com/bycnfz/weather2k.

**Electricity** (Deng et al., 2021): It is publicly available at https://github.com/JLDeng/ST-Norm.

## F  THEORETICAL PROOFS

### F.1  PROOF FOR **PROPOSITION 4.1**

*Proof.* We firstly provide the proof for $I(X; Y \mid Z) = H(X, Z) + H(Y, Z) - H(X, Y, Z) - H(Z)$.

By utilizing the definition of conditional mutual information, $I(X; Y \mid Z)$ can be expressed as follows:

$$I(X;Y \mid Z) = H(X \mid Z) + H(Y \mid Z) - H(X,Y \mid Z) \tag{21}$$

By expanding each term using the definition of conditional entropy, we can obtain:

$$H(X \mid Z) = H(X,Z) - H(Z) \tag{22}$$

$$H(Y \mid Z) = H(Y,Z) - H(Z) \tag{23}$$

$$H(X,Y \mid Z) = H(X,Y,Z) - H(Z) \tag{24}$$

Then we have:

$$I(X;Y \mid Z) = (H(X,Z) - H(Z)) + (H(Y,Z) - H(Z)) - (H(X,Y,Z) - H(Z)) \tag{25}$$

Simplifying the equation, we can obtain:

$$I(X;Y \mid Z) = H(X,Z) + H(Y,Z) - H(X,Y,Z) - H(Z) \tag{26}$$

Proofs of $I(X;Y \mid Z) = H(X,Z) + H(Y,Z) - H(X,Y,Z) - H(Z)$ have been completed. Then, we have the following equivalent expression:

$$\begin{aligned}
I(X;Y \mid Z) &= H(X,Z) + H(Y,Z) - H(X,Y,Z) - H(Z) \\
&= [H(X) + H(Y) - H(X,Y)] \\
&\quad - [H(Z) + H(X,Y) - H(X,Y,Z)] \\
&\quad + [H(Z,Y) + H(X,Y) - H(X,Y,Z) - H(Y)] \\
&\quad + [H(Z,X) + H(Y,X) - H(X,Y,Z) - H(X)]
\end{aligned} \tag{27}$$

By using the following definitions:

$$I(X;Y) = H(X) + H(Y) - H(X,Y), \tag{28}$$
$$I(Z;X,Y) = H(Z) + H(X,Y) - H(X,Y,Z), \tag{29}$$
$$I(Z;X \mid Y) = H(Z,Y) + H(X,Y) - H(X,Y,Z) - H(Y), \tag{30}$$
$$I(Z;Y \mid X) = H(Z,X) + H(Y,X) - H(X,Y,Z) - H(X), \tag{31}$$

We have:

$$I(X;Y \mid Z) = I(X;Y) - (I(Z;X,Y) - I(Z;X \mid Y) - I(Z;Y \mid X)) \tag{32}$$

According to **Definition B.8**, we draw the conclusion as follows:

$$I(Z;Y \mid X) + I(Z;X \mid Y) = I(Z;X,Y) - I(X;Y;Z) \tag{33}$$

### F.2 PROOF FOR **PROPOSITION 4.2**

*Proof.* Consider the mutual information $I(\tilde{X};X)$ defined as follows:

$$I(\tilde{X};X) = \mathbb{E}_{\tilde{X},X}\left[\log\left(\frac{P(\tilde{X}|X)}{P(\tilde{X})}\right)\right]. \tag{34}$$

We parameterize the conditional distribution $P(\tilde{X}|X)$ by utilizing $P_{\phi_x}(\tilde{X}|X)$, and substituting the marginal distribution $P(\tilde{X})$ with a variational approximation $Q(\tilde{X})$, which introduces an extra $KL(P(\tilde{X})\|Q(\tilde{X}))$ term, we get:

$$I(\tilde{X};X) = \mathbb{E}_{\tilde{X},X}\left[\log\left(\frac{P_{\phi_x}(\tilde{X}|X)}{Q(\tilde{X})}\right)\right] - KL(P(\tilde{X})\|Q(\tilde{X})). \tag{35}$$

Using the non-negativity of the Kullback-Leibler divergence, we establish an upper bound:

$$I(\tilde{X}; X) \leq \mathbb{E}\left[KL(P_{\phi_x}(\tilde{X}|X)\|Q(\tilde{X}))\right]. \tag{36}$$

Similarly, for the mutual information $I(\tilde{Y}; Y)$, we have:

$$I(\tilde{Y}; Y) \leq \mathbb{E}\left[KL(P_{\phi_y}(\tilde{Y}|Y)\|Q(\tilde{Y}))\right]. \tag{37}$$

### F.3 PROOF FOR **PROPOSITION 4.3**

*Proof.* We demonstrate the proofs by utilizing $\mathcal{L}_x$ as an example, which is the upper bound of the input regularization. Considering $\mathcal{L}_x$ as the Kullback-Leibler divergence from a normal distribution $\mathcal{N}(\mu_x, \sigma_x^2)$ to the standard normal distribution $\mathcal{N}(0, 1)$, the divergence is given by:

$$
\begin{aligned}
\mathcal{L}_x &= KL\left(\mathcal{N}(\mu_x, \sigma_x^2) \,\|\, \mathcal{N}(0, 1)\right) \\
&= \int \frac{1}{\sqrt{2\pi\sigma_x^2}} \exp\left(-\frac{(x-\mu_x)^2}{2\sigma_x^2}\right) \log\left(\frac{\exp\left(-\frac{(x-\mu_x)^2}{2\sigma_x^2}\right)}{\sqrt{2\pi\sigma_x^2}\exp\left(-\frac{x^2}{2}\right)}\right) dx \\
&= \int \frac{1}{\sqrt{2\pi\sigma_x^2}} \exp\left(-\frac{(x-\mu_x)^2}{2\sigma_x^2}\right) \left[-\frac{1}{2}\log(2\pi\sigma_x^2) - \frac{(x-\mu_x)^2}{2\sigma_x^2} + \frac{x^2}{2}\right] dx \\
&= \frac{1}{2}\left[-\log(\sigma_x^2) + 1 - \frac{1}{\sigma_x^2}\int \exp\left(-\frac{(x-\mu_x)^2}{2\sigma_x^2}\right)(x-\mu_x)^2\, dx + \int \exp\left(-\frac{(x-\mu_x)^2}{2\sigma_x^2}\right) x^2\, dx\right] \\
&= \frac{1}{2}\left[-\log(\sigma_x^2) + \sigma_x^2 + \mu_x^2 - 1\right].
\end{aligned}
\tag{38}
$$

Analogously, the upper bound of the target regularization, denoted as $\mathcal{L}_y$, can be similarly derived and results in:

$$\mathcal{L}_y = \frac{1}{2}\left(-\log\sigma_y^2 + \sigma_y^2 + \mu_y^2 - 1\right). \tag{39}$$

### F.4 PROOF FOR **PROPOSITION 4.4**

*Proof.* The proof can be found in the similar *Proof.* F.2 and *Proof.* F.3.

### F.5 PROOF FOR **PROPOSITION 4.5**

*Proof.* Without holding the Markov chain condition $Z - X - Y$, we cannot derive the lower bound of $I(Z; Y)$ in (Alemi et al., 2016). Therefore, We re-derive the substituted lower bound of $I(Z; Y)$ with the additional term $I(Z; Y|X)$ that arises upon relaxing the constraint $Z - X - Y$. Then, we establish the lower bound for our objective.

$$
\begin{aligned}
I(Z;\tilde{Y}) &= KL\left(\int P(Z|\tilde{Y},X)P(\tilde{Y},X)\,dx \,\big\|\, P(Z)P(\tilde{Y})\right) \\
&= \int P(Z|X,\tilde{Y})P(X,\tilde{Y})\log\frac{P(\tilde{Y}|Z)P(Z)}{P(Z)P(\tilde{Y})}\,dz\,dx\,d\tilde{y} \\
&= \mathbb{E}_{P(X,\tilde{Y})}\left[\int P(Z|X,\tilde{Y})\log P(\tilde{Y}|Z)\,dz\right] \\
&\quad - \mathbb{E}_{P(X,\tilde{Y})}\left[\log P(\tilde{Y})\int P(Z|X,\tilde{Y})\,dz\right] \\
&= \mathbb{E}_{P(X,\tilde{Y})}\mathbb{E}_{P(Z|X,\tilde{Y})}[\log P(\tilde{Y}|Z)] + H(\tilde{Y}) \\
&= \mathbb{E}_{P(X)}\mathbb{E}_{P(\tilde{Y}|X)}\mathbb{E}_{P(Z|X,\tilde{Y})}[\log P(\tilde{Y}|Z,X)] + H(\tilde{Y}) \\
&= \mathbb{E}_{P(X)}\int\int P(Z,\tilde{Y}|X)\log P(Z,\tilde{Y}|X)\,dz\,d\tilde{y} + H(\tilde{Y}) \\
&= \mathbb{E}_{P(X)}\int\int P(Z,\tilde{Y}|X)\log\frac{P(\tilde{Y}|X)P(Z,\tilde{Y}|X)}{P(\tilde{Y}|X)P(Z|X)}\,dz\,d\tilde{y} + H(\tilde{Y}) \\
&= \mathbb{E}_{P(X)}[KL\left(P(\tilde{Y},Z|X)\,\big\|\,P(\tilde{Y}|X)P(Z|X)\right) + \int\int P(Z,\tilde{Y}|X)\log P(\tilde{Y}|X)\,dz\,d\tilde{y}] + H(\tilde{Y}) \\
&= \mathbb{E}_{P(X)}[KL\left(P(\tilde{Y},Z|X)\,\big\|\,P(\tilde{Y}|X)P(Z|X)\right) + \int P(\tilde{Y}|X)\log P(\tilde{Y}|X)\,d\tilde{y}] + H(\tilde{Y}) \\
&= \mathbb{E}_{P(X)}[KL\left(P(\tilde{Y},Z|X)\,\big\|\,P(\tilde{Y}|X)P(Z|X)\right) + \int\int P(\tilde{Y}|X)P(Z|X)\log P(Z|X)\,dz\,d\tilde{y}] + H(\tilde{Y}) \\
&= \mathbb{E}_{P(X)}[KL\left(P(\tilde{Y},Z|X)\,\big\|\,P(\tilde{Y}|X)P(Z|X)\right) \\
&\quad + \int\int P(\tilde{Y}|X)P(Z|X)\log\frac{P(Z|X)P(\tilde{Y}|X)P(Z,\tilde{Y}|X)}{P(Z,\tilde{Y}|X)P(Z|X)}\,dz\,d\tilde{y}] + H(\tilde{Y}) \\
&= \mathbb{E}_{P(X)}[KL\left(P(\tilde{Y},Z|X)\,\big\|\,P(\tilde{Y}|X)P(Z|X)\right) + KL\left(P(\tilde{Y}|X)P(Z|X)\,\big\|\,P(\tilde{Y},Z|X)\right)] + H(\tilde{Y}) \\
&= \mathbb{E}_{P(X)}[KL\left(P(\tilde{Y},Z|X)\,\big\|\,P(\tilde{Y}|X)P(Z|X)\right) + KL\left(P(\tilde{Y}|X)P(Z|X)\,\big\|\,P(\tilde{Y},Z|X)\right) \\
&\quad + \mathbb{E}_{P(Z|X)P(\tilde{Y}|X)}log(P(\tilde{Y}|Z,X))] + H(\tilde{Y}) \\
&= I(\tilde{Y};Z|X) + LI(\tilde{Y};Z|X) + \mathbb{E}_{P(X)}\mathbb{E}_{P(Z|X)P(\tilde{Y}|X)}\log P(\tilde{Y}|Z) + H(\tilde{Y}) \\
&\geq \mathbb{E}_{P(X)}\mathbb{E}_{P(Z|X)P(\tilde{Y}|X)}\log P(\tilde{Y}|Z) + H(\tilde{Y})
\end{aligned}
\tag{40}
$$

Let $Q(\tilde{Y}|Z)$ be the variational approximation of the intractable $P(\tilde{Y}|Z)$, similar to Eq.(35). By the non-negativity of the KL divergence, we have:

$$
KL(P(\tilde{Y}|Z) \,\|\, Q(\tilde{Y}|Z)) \geq 0. \tag{41}
$$

Thus, the inequality simplifies to:

$$
\begin{aligned}
I(Z;\tilde{Y}) &\geq \mathbb{E}_{P(X)}\mathbb{E}_{P(Z|X)P(\tilde{Y}|X)}\log P(\tilde{Y}|Z) + H(\tilde{Y}) \\
&\geq \mathbb{E}_{P(X)}\mathbb{E}_{P(Z|X)P(\tilde{Y}|X)}\log Q(\tilde{Y}|Z) + H(\tilde{Y}).
\end{aligned}
\tag{42}
$$

Since the entropy $H(\tilde{Y})$ is independent of the optimization, we can maximize $I(Z,\tilde{Y})$ by maximizing $\mathbb{E}_{P(X)}\mathbb{E}_{P(Z|X)P(\tilde{Y}|X)}\log Q(\tilde{Y}|Z) \approx -\mathcal{L}_{reg}(Y^S,\tilde{Y})$, where $Y^S$ represents the predictive outputs of RSTIB-MLP model.

## G    SANITY CHECK FOR RSTIB-MLP

In this section, we perform a sanity check on the RSTIB-MLP model to determine whether the instantiation impairs the Information Bottleneck(IB) nature. By conducting this analysis, we aim to theoretically ensure that the RSTIB principle, as an extension of the IB, does not reduce to undesirable degenerate solutions.

**Table 6: Comparison of Assumed Markov Chains, Structural Equations, and Corresponding Directed Acyclic Graphs (DAGs)**

| | | |
|---|---|---|
| **Assumed Markov chain** | $Z - X - Y$ | $X - Z - Y$ |
| **Possible set of structural equations** | $Z = f_Z(X, \eta_Z)$ $Y = f_Y(X, \eta_Y)$ | $Z = f_Z(X, \eta_Z)$ $Y = f_Y(Z, \eta_Y)$ |
| **Corresponding DAG** | $X \longrightarrow Y$ $\searrow$ $Z$ | $X \rightarrow Z \rightarrow Y$ |

As articulated by (Wieczorek & Roth, 2020), the assumptions underlying different Information Bottleneck (IB) principles correspond to different admissible information flows, which can be effectively represented using Directed Acyclic Graphs (DAGs). This approach allows for a convenient elucidation of the properties in different IB models. The arrows in the DAGs explicitly symbolize the data generation process rigorously defined by a corresponding set of equations.

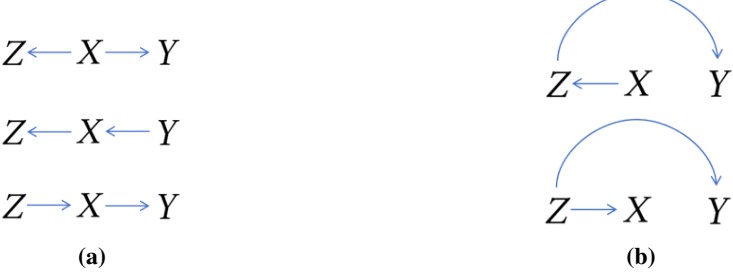

$$Z \longleftarrow X \longrightarrow Y$$
$$Z \longleftarrow X \longleftarrow Y$$
$$Z \longrightarrow X \longrightarrow Y$$
(a)

$$Z \longleftarrow X \quad Y$$
$$Z \longrightarrow X \quad Y$$
(b)

**Figure 10: Admissible DAGs Under Different Markov Assumptions while not impairing IB nature. (a)** $Z - X - Y$; **(b)** $X - Z - Y$.

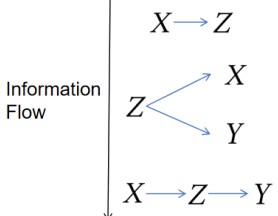

**Figure 11: DAGs of RSTIB-MLP.**

As depicted in Fig.10a, the Markov chain assumption $Z - X - Y$ serves as a sufficient condition to preclude the model from deriving the trivial solution $Z = Y$. Nonetheless, the necessary condition is that $Z$ should not directly depend on $Y$. Consequently, the requirement $Z - X - Y$ can be relaxed to merely prohibiting a direct edge from $Y$ to $Z$ in the DAGs, *i.e.*, $Y \to Z$. This relaxation is achieved by adhering to the admissible DAGs under the Markov assumption $X - Z - Y$, as depicted in Fig.10b. Moreover, $Z$ must encapsulate information about both $X$ and $Y$, necessitating the exclusion of structures $Z \to X \leftarrow Y$ and $Z \to Y \leftarrow X$ in the DAGs, which would otherwise result in $I(Z; Y) = 0$ and $I(X; Z) = 0$, respectively. To summarize, since we also lift the Markov restriction

$Z - X - Y$ by just holding $X - Z - Y$ condition, it is imperative to adhere to the DAGs outlined in Fig. 10b. This mandates a thorough sanity check of the RSTIB-MLP model.

The DAGs of RSTIB-MLP are presented in Fig. 11. As is shown in the figure, our model effectively ensures that it does not reduce to the solution $Z = Y$ while simultaneously guaranteeing the preservation of information from both $X$ and $Y$, thereby maintaining the nature of the information bottleneck principle.

# H  COMPUTATIONAL COMPLEXITY ANALYSIS

In this analysis, we theoretically compare the computational complexities of our RSTIB-MLP with other leading baselines in spatial-temporal forecasting. Many advanced STGNN-based methods integrate Temporal Convolutional Networks (TCNs) and Graph Convolutional Networks (GCNs) with self-attention mechanisms to effectively capture temporal and spatial dependencies, respectively. In contrast, our RSTIB-MLP employs Multi-Layer Perceptrons (MLPs) alone, simplifying the model architecture. This section provides a detailed analysis of the computational complexity associated with these fundamental model architectures.

**Table 7: Notation for Computational Complexity Analysis of GCNs and Self-Attention Mechanisms.**

| Symbol | Description |
|---|---|
| $N$ | The number of time series |
| $\mathcal{E}$ | The edge matrix |
| $\|\mathcal{E}\|$ | The number of edges |
| $A$ | The adjacency matrix, where $A \in \mathbb{R}^{N \times N}$ |
| $d$ | The hidden dimension of each time series |
| $\overline{deg}$ | The average degree of the time series |
| $\hat{A}$ | The adjacency matrix with self-loops, $\hat{A} = A + I$ where $I$ is the identity matrix |
| $\hat{\mathcal{D}}$ | The diagonal degree matrix corresponding to $\hat{A}$, where $\hat{\mathcal{D}}_{ii} = \sum_j \hat{A}_{ij}$ |
| $\hat{A}'$ | The normalized adjacency matrix, $\hat{A}' = \hat{\mathcal{D}}^{-\frac{1}{2}} \hat{A} \hat{\mathcal{D}}^{-\frac{1}{2}}$ |
| $Z$ | The feature matrix, where $Z \in \mathbb{R}^{N \times d}$ |
| $W^{(l)}$ | The feature transformation matrix for the $l$-th layer, $\in \mathbb{R}^{d \times d}$ |
| $\tau(\cdot)$ | A non-linear activation function |
| $L$ | The total number of layers in the network |
| $W_Q^{(l)}$ | The query matrix for the $l$-th layer of the self-attention mechanism, $W_Q^{(l)} \in \mathbb{R}^{d \times d}$ |
| $W_K^{(l)}$ | The key matrix for the $l$-th layer of the self-attention mechanism, $W_K^{(l)} \in \mathbb{R}^{d \times d}$ |
| $W_V^{(l)}$ | The value matrix for the $l$-th layer of the self-attention mechanism, $W_V^{(l)} \in \mathbb{R}^{d \times d}$ |

**Computational Complexity of GCN.** We detail the computational complexities of GCNs based on the notations provided in Table 7. The computation at the $l$-th layer of a GCN can be expressed as:

$$Z^{(l+1)} = \tau(\hat{A}' Z^{(l)} W^{(l)}) \tag{43}$$

which can typically be divided into two primary operations:

- **Feature Transformation**: $Z'^{(l)} = Z^{(l)} W^{(l)}$.

- **Neighborhood Aggregation**: $Z^{(l+1)} = \tau(\hat{A}' Z'^{(l)})$.

Thus, naively, the computational complexity of GCN can be expressed as:

$$O(L \cdot (N \cdot d^2 + N^2 \cdot d)) \tag{44}$$

In practice, the scatter function from Pytorch (Paszke et al., 2019) can efficiently handle the graph structure's sparsity. Given that the average degree of nodes is denoted by $\overline{deg}$, the complexity for

neighborhood aggregation per node is $O(\overline{deg} \times d)$, resulting in a total of $O(N \times \overline{deg} \times d) = O(|\mathcal{E}| \times d)$. Thus, the practical computational complexity of a GCN is:

$$O(L \cdot (N \cdot d^2 + |\mathcal{E}| \cdot d)) \tag{45}$$

Generally, the complexity of the activation function $\tau(\cdot)$, being an element-wise operation, is negligible and can be approximated as $O(N)$.

When combining GCNs with a Self-Attention mechanism, the query, key, and value matrices in the $l$-th layer, denoted as $W_Q^l$, $W_K^l$, and $W_V^l$ respectively, are all $d \times d$ matrices. The self-attention mechanism involves the following computations:

1. Compute $Q^{(l)} = Z^{(l)} W_Q^{(l)}$, $K^{(l)} = Z^{(l)} W_K^{(l)}$, and $V^{(l)} = Z^{(l)} W_V^{(l)}$, each with a computational cost of $O(Nd^2)$.

2. Compute the product $Q^{(l)} K^{(l)\top}$, which incurs a cost of $O(N^2 d)$.

3. Compute the final attention scores, requiring $O(N^2 d)$ time.

Therefore, the total computational complexity when incorporating self-attention is:

$$O(L \cdot (N^2 d + Nd^2)) \tag{46}$$

**Computational Complexity of TCNs.** We detail the computational complexities of TCNs based on the notations provided in Table 8. TCNs integrated with attention mechanisms are often benchmarked against sequential models such as RNNs and LSTMs. The computational complexity for these sequence models is typically $O(L \times T \times N^2 \times d^2)$. However, similar to the above analysis, TCNs equipped with attention mechanisms generally incur lower computational costs, estimated at $O(L \times N \times T^2 \times d)$. The reduced complexity is attributed to the faster learning dynamics of $T^2$ compared to $(N \times d)^2$. Although TCNs have been demonstrated to enhance efficiency significantly (Zhou et al., 2020), they are still considered sub-optimal compared to MLP networks.

**Table 8: Notation for Computational Complexity Analysis of TCNs with Attention Mechanisms.**

| Symbol | Description |
|---|---|
| $L$ | The number of layers in the model |
| $T$ | The length of the time series |
| $N$ | The number of time series |
| $d$ | The hidden dimension |

**Computational Complexity of RSTIB-MLP Networks.** The RSTIB-MLP architecture employs a straightforward encoder-decoder MLP network design. We denote $d_{in}$ as the input dimension, $d_{out}$ as the output dimension, and $d$ as the dimension of the hidden layer. The computational complexity of the model can be succinctly expressed as $O(N \times (d_{in} \times d + d_{out} \times d))$, where $N$ represents the number of time series being processed.

**Table 9: Notation for Computational Complexity Analysis of RSTIB-MLP Networks.**

| Symbol | Description |
|---|---|
| $N$ | The number of time series |
| $d_{in}$ | The dimension of the input |
| $d_{out}$ | The dimension of the output |
| $d$ | The dimension of the hidden layer in the MLP network |

Thus, theoretically, RSTIB-MLP's computational complexity is considerably more efficient than that of STGNN-based methods, primarily due to its streamlined MLP-based model architecture.

# I  FURTHER DISCUSSIONS

## I.1  LIMITATIONS

Our general framework leaves many interesting questions for future investigation. For example, could we automatically search for better regularization coefficients with theoretical and empirical efficiency

guarantees? Besides, MLPs with specially designed modules have been proven to be effective. Could we instantiate the RSTIB principle to incorporate more reliable spatial-temporal information from the module design? These are all the limitations and future directions that we are attempting to explore.

## I.2 A FURTHER COMPARISON OF RSTIB AND OTHER IB PRINCIPLES

This section elaborates on the comparative analysis between the Information Bottleneck(IB), its variants and our proposed Robust Spatial-Temporal Information Bottleneck (RSTIB) principle. Notably, RSTIB extends the capabilities of Deep Variational Information Bottleneck (DVIB) and Robust Graph Information Bottleneck (RGIB). Given that RGIB itself generalizes the Graph Information Bottleneck (GIB), our comparison primarily focuses on the IB model as introduced by (Tishby et al., 2000), alongside its significant extensions: DVIB (Alemi et al., 2016), GIB (Wu et al., 2020), RGIB (Zhou et al., 2023), and our RSTIB.

Briefly speaking, compared to IB, as well as DVIB and GIB derivatives, the RSTIB introduces significant advancements by accounting for spatial-temporal data noise present both in input and target regions, enhancing robustness in both theoretical constructs and instantiation. While the RSTIB extends the RGIB principle, it diverges by considering lifting specific Markov assumption typically held to explicitly minimize the irrelevant information terms. The subsequent reformulations can ensure the integrity of the IB principle. In other words, RSTIB ensures, both theoretically and practically, that encoding $Z$ does not reduce to the trivial solution $Z = Y$ and preserves information from both $X$ and $Y$. This enhancement is meticulously analyzed in Section G, a thoroughness not typically found in RGIB's analysis. Furthermore, the instantiation of the RSTIB principle does not depend on specific data structural assumptions inherent to the instantiations of RGIB, which are based on graph data and assume that the number of edges in the pruned graph, denoted as $|Z_A|$, does not exceed those in the original graph, $|A|$. Therefore, the RSTIB framework demonstrates more general potential applications and robustness, making it suitable for instantiating in Multi-Layer Perceptron (MLP) networks for spatial-temporal forecasting.

Analytically, traditional models such as IB, DVIB, and GIB predominantly focus on minimizing the conditional entropy $H(X|Y)$ while maximally preserving $H(Y|X)$. These models operate under the implicit assumption that $I(Z;Y|X) = 0$, adhering strictly to the Markov chain condition $Z - X - Y$. This approach proves effective for specific applications, such as classification tasks. However, in spatial-temporal forecasting, with the Assumption 4.1 and 4.2 proposed about spatial-temporal data, such Markov assumption is too restrictive. The noise-related irrelevant information could be obscured within this restriction, thereby questioning the direct adoption of the Markov assumption $Z - X - Y$. Besides, RGIB, by its definition, considers an explicit relationship between the information terms and attempts to balance them in a self-controlled way. Some of its derived terms, such as $H(Z|X,Y)$, is minimized by controlling $H(Z)$ to be within the range $\gamma_H^- < H(Z) < \gamma_H^+$, given that $H(Z) \geq \max\{H(Z|X), H(Z|Y)\} \geq H(Z|X,Y)$. This requires a delicate balance within the RGIB objective. In comparison to these, RSTIB adopts a distinct approach. It lifts the Markov condition of $Z - X - Y$ by adhering to only the $X - Z - Y$ assumption, which is less restrictive while not impairing the bottleneck nature of the representation $Z$. This formulation introduces $I(Z;Y|X) \neq 0$, with the existing $I(Z;X|Y)$ to be minimized, enhancing robustness against noise perturbations in both input and target. Besides, RSTIB focuses on learning the "*minimal sufficient representation*" while minimizing explicitly expressed and reformulated irrelevant information under the $X - Z - Y$ Markov assumption. This strategic orientation provides a theoretical guarantee that the encoding of $Z$ neither reduces to the trivial solution $Z = Y$ nor compromises the information from $X$ and $Y$ which results in $I(X;Z) = 0$ and $I(Z;Y) = 0$ respectively.

Regarding instantiations, the GIB is inherently intertwined with the Graph Attention Network (GAT) architecture. While the RGIB mitigates this constraint by eliminating the need to modify the Graph Neural Network (GNN) architecture, it still necessitates reliance on GNNs and their inherent graph structures. This dependence is under the assumption that the number of edges in the pruned graph, denoted as $|Z_A|$, does not exceed those in the original graph, denoted as $|A|$. However, such an assumption can not hold when generalizing to Multi-Layer Perceptron (MLP) networks, where graph structures are inapplicable. Meanwhile, spatial-temporal data often comes with no pre-defined graph structure. GIB/RGIB-based method can not directly be applied to such scenario. Besides, the DVIB adheres strictly to both Markov chain assumptions, which imposes overly restrictive constraints on the optimization process for the potential set of joint distributions $P(X, Y, Z)$. In response to these

limitations, we propose the RSTIB-MLP instantiation, as outlined in Section 4.2. The information flow of RSTIB-MLP follows Fig.11, which is under less restrictive Markov assumption for a wider array of potential joint distributions while not impairing IB nature with theoretical guarantees. Besides, it integrates three independent and identically distributed (i.i.d.) Gaussian distributions as the prior distributions for the input, representation, and target regions, which play the role of minimizing the irrelevant information during the optimization. These circumvent the dependency on graph structures, which can be instantiated in MLPs for spatial-temporal forecasting.

# J EXPERIMENTAL IMPLEMENTATION DETAILS

We provide a comprehensive description of the experimental settings. For all experiments, the best models are selected based on the Mean Absolute Error (MAE) metric on the validation set. All comparative baselines are trained using their default settings. The models are trained on NVIDIA GeForce RTX 3090Ti GPUs, utilizing the PyTorch framework (Paszke et al., 2019). The main code bases referenced are STID (Shao et al., 2022a) and STExplainer (Tang et al., 2023), as implemented in https://github.com/zezhishao/STID and https://github.com/HKUDS/STExplainer, respectively. Besides, regarding the attachment of spatial-temporal information, we adopt the spatial-temporal prompts technique in STGKD (Tang et al., 2024)(https://openreview.net/forum?id=akKNGGWegr). It combines spatial-temporal identity (Shao et al., 2022a) with dynamic graph construction (Han et al., 2021). The reference implementation for this technique can be found in https://github.com/zezhishao/STID and https://github.com/liangzhehan/DMSTGCN. We implement the noise injection by firstly loading the original datasets, then we conduct the data normalization. Further, we build the index information about time of day and day of week. Notably, The index information is not perturbed by the noise. It is concatenated with perturbed input afterwards. We implement this attachment by adopting the default settings in their works and combining them following the guideline of (Tang et al., 2024) for fair comparison. The additional hyperparameter settings and additional experimental details are provided in the subsequent sub-sections.

## J.1 IMPLEMENTATION DETAILS FOR RSTIB-MLP

We adopt PyTorch 1.13.1 on NVIDIA RTX 3090Ti GPUs. The algorithm of RSTIB-MLP is shown in **Algorithm 1**. We follow STID (Shao et al., 2022a)'s default model configuration, using 3 Multi-Layer Perceptrons layers. The nonlinear activation $\tau$ is ReLU. We follow STID's default learning rate setting, *i.e.*, we initialize the learning rate $\eta = 0.002$, and apply a decay factor $r = 0.5$ for all three benchmarks. A summary of the default hyperparameter settings is in Table 10. Table 10 provides the hyperparameters that produce the results in Section 5.1. For some specific hyperparameters with a searching space, we provide the results of hyperparameter investigation, mainly consisted of the Lagrange multipliers $\lambda_x \in \{0.01, 0.001, 0.0001, 0.00001\}$, $\lambda_y \in \{0.01, 0.001, 0.0001, 0.00001\}$, $\lambda_z \in \{0.01, 0.001, 0.0001, 0.00001\}$, and the distance function $D \in \{MAE, SmoothL1, MSE\}$, evaluated in PEMS04 dataset in Section K.9.

## J.2 ADDITIONAL DETAILS FOR ROBUSTNESS STUDY

Each specified data noise ratio is termed as $\gamma$. And we perform random spatial-temporal noise perturbation by adding independent Gaussian noise $\gamma \cdot \epsilon$ to each feature dimension of the time series, where $\epsilon \sim N(0, 1)$.

**Table 10: Hyperparameter scope for Section 5.1**

| Hyperparameters | Value/Search space | Type |
|---|---|---|
| Batch Size $B$ | **32** | Fixed* |
| Epoch $E$ | **200** | Fixed |
| Learning Rate $\eta$ | **0.002** | Fixed |
| Decay Factor $r$ | **0.5** | Fixed |
| Hidden Dimension $d$ | **64** | Fixed |
| Number of MLP Layers $L$ | **3** | Fixed |
| Non-Linear Activation $\tau$ | **ReLU** | Fixed |
| Input Regularization Coefficient $\lambda_x$ | $\{0.01, \mathbf{0.001}, 0.0001, 0.00001\}$ | Choice† |
| Target Regularization Coefficient $\lambda_y$ | $\{0.01, \mathbf{0.001}, 0.0001, 0.00001\}$ | Choice |
| Representation Regularization Coefficient $\lambda_z$ | $\{0.01, \mathbf{0.001}, 0.0001, 0.00001\}$ | Choice |
| Distance Function $D$ | $\{\mathbf{MAE}, SmoothL1, MSE\}$ | Choice |

*Fixed: a constant value

†Choice: choose from a set of discrete values

The **boldface** numbers: Default setting that produces the result for Section 5.1

## K   FURTHER EMPIRICAL RESULTS

### K.1   ADDITIONAL PERFORMANCE COMPARISON ON CLEAN PEMS DATASETS

In this section, we provide additional empirical study for the comparison between the performance of RSTIB-MLP and more baselines targeting on spatial-temporal traffic forecasting. The results are shown in Table 11. By examing the results, it's more convincing that our predictive performance when learning with clean data can be superior, even when comparing with STGNNs.

**Table 11: Performance Comparison Under Clean PEMS04, PEMS07, PEMS08 Datasets. The boldface means the best results.**

| Dataset | Metrics | Methods | | | | | | | |
|---|---|---|---|---|---|---|---|---|---|
| | | STGCN | AGCRN | GMSDR | FOGS | DSTAGNN | STHMLP | TrendGCN | **RSTIB-MLP** |
| | MAE | 20.05 | 19.83 | 20.49 | 19.74 | 19.30 | 18.88 | 18.81 | **18.46** |
| PEMS04 | RMSE | 32.07 | 32.26 | 32.13 | 31.66 | 31.46 | 30.31 | 30.68 | **30.14** |
| | MAPE(%) | 13.09 | 12.97 | 14.15 | 13.05 | 12.70 | 12.74 | 12.25 | **12.22** |
| | MAE | 21.98 | 22.37 | 22.27 | 21.28 | 21.42 | 20.71 | 20.43 | **19.84** |
| PEMS07 | RMSE | 35.66 | 36.55 | 34.94 | 34.88 | 34.51 | 33.99 | 34.32 | **33.90** |
| | MAPE(%) | 9.28 | 9.12 | 9.86 | 8.95 | 9.00 | 8.75 | 8.51 | **8.33** |
| | MAE | 16.39 | 15.95 | 16.36 | 15.73 | 15.67 | 15.22 | 15.15 | **14.51** |
| PEMS08 | RMSE | 25.60 | 25.22 | 25.58 | 24.92 | 24.77 | **24.18** | 24.26 | **24.18** |
| | MAPE(%) | 10.34 | 10.09 | 10.28 | 9.88 | 9.94 | 9.82 | 9.51 | **9.44** |

### K.2   FURTHER ABLATION STUDY ON EACH REGULARIZATIONS

**Table 12: Performance of RSTIB-MLP under different noise ratios and ablated regularization on PEMS04**

| Noise Ratio | 10% | | | 30% | | | 50% | | |
|---|---|---|---|---|---|---|---|---|---|
| Method | MAE | RMSE | MAPE | MAE | RMSE | MAPE | MAE | RMSE | MAPE |
| RSTIB-MLP w/o x+y+z | 24.97 | 37.67 | 16.55% | 29.64 | 45.75 | 20.43% | 29.80 | 46.71 | 19.23% |
| RSTIB-MLP w/o x+y | 24.50 | 37.28 | 16.23% | 28.84 | 45.12 | 17.91% | 29.19 | 45.87 | 18.79% |
| RSTIB-MLP w/o y | 23.81 | 36.60 | 15.56% | 27.49 | 43.58 | 16.72% | 27.98 | 44.48 | 17.85% |
| RSTIB-MLP | 23.64 | 36.44 | 15.22% | 27.15 | 42.85 | 17.19% | 27.16 | 43.43 | 17.76% |

In this section, we aim to separately examine each regularization term within the objective function, including input, target, and representation regularizations. For simplicity, we denote "w/o" meaning the word "without", "x" as input regularization, "y" as target regularization and "z" as representation regularization respectively. As

is shown in Table 12, all the regularization terms can enhance the performance. Notably, there is a fact that the input regularization contributes significantly compared with other regularization terms, while the contributions from the representation and target regularization terms are comparable. The potential reason may lies on the fact that applying input regularization ensures that the signals passed to subsequent layers are less noisy. Besides, input regularization can prevent the model from overly relying on specific input patterns, thereby enhancing robustness. In contrast, solely regularizing the representation may not sufficiently address the complexity and noise present in the input data.

## K.3 FURTHER EMPIRICAL STUDY OF COMBATING DATA MISSING

To demonstrate broader applicability, we conduct a performance comparison and an ablation study showcasing how each module performs when combating against noise arising from data missing. We conduct this experiment on the PEMS04 dataset, where we randomly drop the data by certain ratios. The results are shown in Table 13 and Table 14.

**Table 13: Performance comparison of different methods under varying missing ratios.**

| Missing Ratio | 10% | | | 30% | | |
|---|---|---|---|---|---|---|
| Method | MAE | RMSE | MAPE | MAE | RMSE | MAPE |
| STG-NCDE | 20.25 | 32.58 | 13.22% | 26.32 | 40.38 | 15.27% |
| STGKD | 20.57 | 32.77 | 13.40% | 29.06 | 44.22 | 16.34% |
| STID | 22.65 | 35.52 | 13.87% | 30.21 | 44.98 | 16.85% |
| RSTIB-MLP | 19.83 | 31.79 | 12.82% | 25.45 | 39.61 | 14.94% |

**Table 14: Ablation Study of RSTIB-MLP modules on the PEMS04 dataset with different missing ratios**

| Missing Ratio | 10% | | | 30% | | |
|---|---|---|---|---|---|---|
| Method | MAE | RMSE | MAPE | MAE | RMSE | MAPE |
| RSTIB-MLP | 19.83 | 31.79 | 12.82% | 25.45 | 39.61 | 14.94% |
| RSTIB-MLP w/o KD | 19.95 | 32.10 | 12.91% | 26.31 | 40.46 | 15.22% |
| RSTIB-MLP w/o KD + RSTIB | 20.57 | 32.77 | 13.40% | 29.06 | 44.22 | 16.34% |
| RSTIB-MLP w/o KD + IB | 21.35 | 33.76 | 13.94% | 29.34 | 44.69 | 17.91% |

These results indicate that RSTIB-MLP can surpass all the MLP-based baselines, even be comparable with STGNNs like STG-NCDE (Choi et al., 2022). Besides, each module also contributes to the overall robustness. Observing from the results, it is obvious that RSTIB implementation can significantly enhance the robustness, especially when combating data missing with higher missing data ratio. Besides, traditional IB implementation and knowledge distillation can also contribute to robustness enhancement, sharing similar conclusions from our previous results.

## K.4 MODEL ARCHITECTURE AGNOSTIC STUDY

To ensure consistency, we implement RSTIB on STID (Shao et al., 2022a) to evaluate whether each module's contribution on enhancing robustness is model- or network architecture-agnostic. Table 15 demonstrates the results conducted on PEMS04. We keep the same notations as in Section 5.2.

**Table 15: Results of RSTIB implementation on STID for clean and noisy PEMS04 datasets.**

| Noise Ratio | 0% (Clean) | | | 10% | | | 30% | | | 50% | | |
|---|---|---|---|---|---|---|---|---|---|---|---|---|
| Metrics | MAE | RMSE | MAPE | MAE | RMSE | MAPE | MAE | RMSE | MAPE | MAE | RMSE | MAPE |
| STID | 18.79 | 30.37 | 12.57% | 27.83 | 41.34 | 17.31% | 36.53 | 52.74 | 21.11% | 36.22 | 52.15 | 21.45% |
| STID+IB | 18.65 | 30.23 | 12.53% | 25.70 | 38.17 | 17.17% | 34.55 | 48.07 | 19.10% | 34.99 | 50.94 | 19.59% |
| STID+RSTIB | 18.57 | 30.16 | 12.51% | 24.27 | 36.89 | 16.34% | 28.67 | 44.73 | 18.06% | 29.02 | 46.44 | 18.37% |
| STID+RSTIB+KD | 18.50 | 30.02 | 12.32% | 23.99 | 36.57 | 16.22% | 28.12 | 44.31 | 17.81% | 28.86 | 45.63 | 17.92% |

The above results demonstrate the consistency of our method's performance:

- Each evaluated module contributes to the enhancement of predictive performance under clean setting. However, we observe that the enhancement of the performance may not be significant. The potential reason could be the fact that our objective function includes more regularization terms, achieving an optimal balance may be harder, leading to potential over-regularization under clean data.

- Each module can also be applied to another baseline model, which is STID (Shao et al., 2022a) in this evaluation. Besides, the enhancement from different modules under noisy scenarios can also be clearly observed. Thus, the RSTIB implementation contributes significantly to the enhancement of STID's ability in combating the noise.

## K.5 THE AVERAGE IMPROVEMENTS OF RSTIB-MLP WHEN COMBATING AGAINST NOISE PERFURBATION

In this section, we calculate the average improvements of RSTIB-MLP when comparing with the best competing methods by averaging over all the noise ratios on each noisy dataset. The results are summarized in Table 16. We can tell from the table that RSTIB-MLP can gain large improvement on several datasets when comparing with specific baselines. For example, RSTIB-MLP improves MAE, RMSE and MAPE by 8.39%, 5.51%, and 3.74% on PEMS04 dataset, and by 7.02%, 4.75%, and 8.08% on PEMS08 dataset compared to one of the best competing methods, STExplainer (Tang et al., 2023).

**Table 16: Average Improvements of RSTIB-MLP Compared with Each Baselines Under Noisy Datasets**

| Noise Ratio | PEMS04 | | | PEMS07 | | | PEMS08 | | |
|---|---|---|---|---|---|---|---|---|---|
| Metrics | MAE | RMSE | MAPE | MAE | RMSE | MAPE | MAE | RMSE | MAPE |
| STID | 33.53(+21.92%) | 48.74(+15.78%) | 19.96(+15.95%) | 30.73(+4.96%) | 52.52(+1.93%) | 13.35(+7.82%) | 24.89(+9.02%) | 42.20(+7.57%) | 15.38(+9.84%) |
| GWN | 36.82(+28.73%) | 51.73(+20.51%) | 21.65(+22.67%) | 32.70(+10.06%) | 53.71(+3.99%) | 17.16(+21.99%) | 25.16(+10.02%) | 42.66(+8.58%) | 15.53(+10.71%) |
| TrendGCN | 26.36(+1.39%) | 41.68(+1.83%) | 19.08(+12.35%) | 31.86(+7.56%) | 52.61(+2.01%) | 19.53(+35.20%) | 24.15(+6.72%) | 39.88(+2.85%) | 20.54(+30.26%) |
| STExplainer-CGIB | 28.64(+8.70%) | 43.85(+6.50%) | 16.96(+4.99%) | 32.73(+10.40%) | 53.61(+3.60%) | 14.39(+13.58%) | 25.01(+10.27%) | 41.08(+6.12%) | 18.4(+20.48%) |
| STExplainer | 28.49(+8.39%) | 43.46(+5.51%) | 17.10(+3.74%) | 30.79(+5.19%) | 51.96(+0.85%) | 14.17(+12.73%) | 24.29(+7.02%) | 40.69(+4.75%) | 15.13(+8.08%) |
| STGKD | 27.37(+4.96%) | 42.69(+3.69%) | 17.53(+4.34%) | 30.28(+3.44%) | 51.96(+0.76%) | 13.33(+7.65%) | 24.06(+6.38%) | 39.92(+2.98%) | 15.87(+12.76%) |
| BiTGraph | 28.74(+9.21%) | 43.78(+6.22%) | 17.24(+4.03%) | 31.05(+5.97%) | 52.31(+1.46%) | 14.29(+13.50%) | 24.57(+8.09%) | 41.00(+5.49%) | 15.27(+8.89%) |
| STC-Dropout | 31.58(+17.32%) | 46.97(+12.48%) | 19.01(+11.73%) | 31.09(+6.07%) | 52.41(+1.65%) | 14.37(+13.94%) | 24.22(+6.82%) | 40.63(+6.04%) | 15.12(+8.32%) |
| STG-NCDE | 28.34(+8.07%) | 42.92(+4.52%) | 19.20(+12.82%) | 31.49(+7.25%) | 53.22(+3.07%) | 15.59(+20.98%) | 26.38(+14.12%) | 40.82(+5.03%) | 16.73(+16.91%) |
| FreTS | 28.70(+9.07%) | 43.72(+6.08%) | 17.22(+3.96%) | 31.32(+6.73%) | 53.01(+2.62%) | 15.47(+20.33%) | 24.54(+7.98%) | 40.97(+5.43%) | 15.23(+8.64%) |
| RSTIB-MLP | 25.98 | 40.91 | 16.72% | 29.21 | 51.55 | 12.31% | 22.48 | 38.70 | 13.85% |
| Noise Ratio | LargeST(SD) | | | Weather2K-R | | | Electricity | | |
| Metrics | MAE | RMSE | MAPE | MAE | RMSE | MAPE | MAE | RMSE | MAPE |
| STID | 32.19(+8.24%) | 49.88(+6.74%) | 19.68(+14.61%) | 6302.23(+7.01%) | 8114.17(+6.21%) | 75.36%(+4.27%) | 38.10(+8.81%) | 64.80(+5.59%) | 31.92%(+9.22%) |
| GWN | 32.53(+9.24%) | 51.61(+9.67%) | 19.96%(+15.61%) | 6808.57(+13.73%) | 8470.20(+10.27%) | 74.69%(+3.38%) | - | - | - |
| TrendGCN | 30.31(+2.80%) | 47.84(+2.98%) | 17.64%(+5.14%) | 5893.37(+0.55%) | 7642.31(+0.56%) | 72.61%(+0.49%) | 35.61(+2.88%) | 62.98(+2.86%) | 29.84%(+3.13%) |
| STExplainer-CGIB | 30.13(+4.75%) | 49.08(+5.38%) | 18.45%(+9.39%) | 6260.13(+6.02%) | 8112.80(+6.01%) | 74.88%(+3.69%) | - | - | - |
| STExplainer | 29.50(+2.86%) | 47.64(+2.58%) | 17.69%(+5.15%) | 6257.63(+5.98%) | 8109.80(+5.97%) | 74.74%(+3.31%) | - | - | - |
| STGKD | 30.43(+2.99%) | 47.92(+2.70%) | 17.57%(+4.76%) | 5891.45(+5.94%) | 7639.89(+5.93%) | 72.48(+3.32%) | 36.35(+4.59%) | 63.82(+4.05%) | 31.07%(+6.38%) |
| BiTGraph | 30.24(+2.75%) | 47.81(+3.30%) | 17.61%(+5.00%) | 5891.45(+0.52%) | 7639.88(+0.53%) | 72.48%(-0.31%) | 36.99(+6.09%) | 63.96(+4.40%) | 31.13%(+7.20%) |
| STC-Dropout | 30.34(+2.88%) | 47.93(+2.83%) | 17.60%(+5.64%) | 6260.31(+6.02%) | 8111.84(+5.99%) | 74.77%(+3.54%) | 37.37(+7.39%) | 64.87(+5.96%) | 31.58%(+9.18%) |
| STG-NCDE | 30.72(+4.12%) | 48.43(+4.38%) | 17.78%(+5.97%) | 6258.56(+6.00%) | 8110.60(+5.98%) | 74.73%(+3.50%) | 37.34(+7.32%) | 64.82(+5.88%) | 31.60%(+9.22%) |
| FreTS | 30.62(+3.79%) | 48.57(+4.43%) | 17.70%(+5.52%) | 5889.14(+0.48%) | 7637.68(+0.50%) | 72.43%(+0.24%) | 38.81(+10.06%) | 65.29(+6.42%) | 32.26%(+10.93%) |
| RSTIB-MLP | 29.47 | 46.42 | 16.73% | 5861.39 | 7599.68 | 72.26% | 34.53 | 60.98 | 28.84% |

## K.6 A STUDY OF AVERAGE PERFORMANCE DECAY COMPARISON

**Table 17: Average Performance Decay Comparison Under Noisy Datasets**

| Noise Ratio | PEMS04 | | | PEMS07 | | | PEMS08 | | |
|---|---|---|---|---|---|---|---|---|---|
| Metrics | MAE | RMSE | MAPE | MAE | RMSE | MAPE | MAE | RMSE | MAPE |
| STID | 33.53(-78.45%) | 48.74(-60.49%) | 19.96%(-59.55%) | 30.73(-50.56%) | 52.52(-55.94%) | 13.35%(-52.75%) | 24.89(-67.38%) | 42.20(-76.05%) | 15.38%(-47.46%) |
| GWN | 36.82(-91.57%) | 51.73(-68.28%) | 21.65%(-72.92%) | 32.70(-61.48%) | 53.71(-61.19%) | 17.16%(-98.84%) | 25.16(-71.51%) | 42.66(-81.61%) | 15.53%(-63.13%) |
| TrendGCN | 26.36(-40.14%) | 41.68(-35.85%) | 19.08%(-55.76%) | 31.86(-55.95%) | 52.61(-53.29%) | 19.53%(-129.49%) | 24.15(-59.41%) | 39.88(-64.39%) | 20.54%(-115.98%) |
| STExplainer-CGIB | 28.64(-49.63%) | 43.85(-42.51%) | 16.96%(-31.37%) | 32.73(-59.27%) | 53.61(-52.65%) | 14.39%(-67.13%) | 25.01(-68.19%) | 41.08(-70.67%) | 18.4%(-79.34%) |
| STExplainer | 28.49(-53.42%) | 43.46(-44.19%) | 17.10%(-40.97%) | 30.79(-53.95%) | 51.96(-55.10%) | 14.17%(-66.71%) | 24.29(-66.48%) | 40.69(-70.18%) | 15.13%(-54.39%) |
| STGKD | 27.37(-46.44%) | 42.69(-40.15%) | 17.53%(-42.06%) | 30.28(-49.16%) | 51.96(-51.49%) | 13.33%(-50.28%) | 24.06(-59.02%) | 39.92(-60.97%) | 15.87%(-48.87%) |
| BiTGraph | 28.74(-52.71%) | 43.78(-43.82%) | 17.24%(-40.73%) | 31.05(-53.33%) | 52.31(-54.99%) | 14.29%(-66.16%) | 24.57(-65.45%) | 41.00(-69.42%) | 15.27%(-54.24%) |
| STC-Dropout | 31.58(-68.43%) | 46.97(-54.61%) | 19.01%(-54.18%) | 31.09(-51.88%) | 52.41(-54.56%) | 14.37%(-64.23%) | 24.22(-64.76%) | 40.63(-67.06%) | 15.12%(-55.08%) |
| STG-NCDE | 28.34(-47.53%) | 42.92(-38.05%) | 19.20%(-50.47%) | 31.49(-53.39%) | 53.22(-57.27%) | 15.59%(-77.16%) | 26.38(-70.74%) | 40.82(-64.53%) | 16.73%(-68.65%) |
| FreTS | 28.70(-52.90%) | 43.72(-43.58%) | 17.22%(-40.57%) | 31.32(-57.23%) | 53.01(-57.53%) | 15.47%(-77.82%) | 24.54(-65.25%) | 40.97(-69.65%) | 15.23%(-53.99%) |
| RSTIB-MLP | 25.98(-40.74%) | 40.91(-35.73%) | 16.72%(-36.82%) | 29.21(-47.23%) | 51.55(-52.06%) | 12.31%(-47.78%) | 22.48(-54.93%) | 38.70(-60.05%) | 13.85%(-46.72%) |
| Noise Ratio | LargeST(SD) | | | Weather2K-R | | | Electricity | | |
| Metrics | MAE | RMSE | MAPE | MAE | RMSE | MAPE | MAE | RMSE | MAPE |
| STID | 32.19(-82.90%) | 49.88(-71.70%) | 19.68(-65.10%) | 6302.23(-57.64%) | 8114.17(-30.88%) | 75.36%(-15.34%) | 38.10(-88.80%) | 64.80(-62.73%) | 31.92%(-100.50%) |
| GWN | 32.53(-83.37%) | 51.61(-74.24%) | 19.96%(-68.01%) | 6808.57(-70.59%) | 8470.20(-36.45%) | 74.69%(-13.17%) | - | - | - |
| TrendGCN | 30.31(-74.30%) | 47.84(-61.46%) | 17.64%(-51.55%) | 5893.37(-47.78%) | 7642.31(-22.80%) | 72.61%(-11.19%) | 35.61(-78.23%) | 62.98(-58.96%) | 29.84%(-89.82%) |
| STExplainer-CGIB | 30.13(-61.99%) | 49.08(-62.03%) | 18.45%(-45.39%) | 6260.13(-56.71%) | 8112.80(-30.83%) | 74.88%(-14.58%) | - | - | - |
| STExplainer | 29.50(-68.48%) | 47.64(-65.07%) | 17.69%(-52.90%) | 6257.63(-56.73%) | 8109.80(-30.84%) | 74.74%(-14.60%) | - | - | - |
| STGKD | 30.43(-72.90%) | 47.92(-62.88%) | 17.57%(-51.20%) | 5891.45(-47.65%) | 7639.89(-23.31%) | 72.48%(-11.37%) | 36.35(-80.40%) | 63.82(-59.35%) | 31.07%(-95.53%) |
| BiTGraph | 30.24(-60.42%) | 47.81(-60.44%) | 17.61%(-38.88%) | 5891.45(-47.68%) | 7639.88(-22.91%) | 72.48%(-11.15%) | 36.99(-85.14%) | 63.96(-60.42%) | 31.13%(-93.11%) |
| STC-Dropout | 30.34(-72.88%) | 47.93(-63.25%) | 17.60%(-50.68%) | 6260.31(-57.04%) | 8111.84(-30.73%) | 74.77%(-14.41%) | 37.37(-87.60%) | 64.87(-62.79%) | 31.58%(-91.74%) |
| STG-NCDE | 30.72(-74.74%) | 48.43(-66.20%) | 17.78%(-49.79%) | 6258.56(-56.76%) | 8110.60(-30.84%) | 74.73%(-14.58%) | 37.34(-88.11%) | 64.82(-62.37%) | 31.60%(-91.28%) |
| FreTS | 30.62(-74.57%) | 48.57(-67.43%) | 17.70%(-48.12%) | 5889.14(-47.81%) | 7637.68(-22.81%) | 72.43%(-11.23%) | 38.81(-92.89%) | 65.29(-61.41%) | 32.26%(-98.89%) |
| RSTIB-MLP | 29.47(-68.40%) | 46.42(-61.46%) | 16.73%(-49.38%) | 5861.39(-47.85%) | 7599.68(-22.75%) | 72.26%(-11.27%) | 34.53(-74.39%) | 60.98(-53.72%) | 28.84%(-83.46%) |

In this section, we aim to investigate if RSTIB-MLP's performance is also superior regarding performance degradation caused by the noise perturbation. The detailed average performance degradation of each baseline, including RSTIB-MLP, by averaging across different noise ratios compared with clean scenario on Table 17. Notably, the performance regarding the average performance decline of RSTIB-MLP is still more superior compared with other baselines. For all the metrics, including MAE, RMSE, MAPE in 6 benchmark datasets($3 \times 6 = 18$) cases), only 3 cases that RSTIB-MLP has not achieved the best or second-best results. Along with the absolute best performance achieved by RSTIB-MLP in all cases, it is still reasonable to claim that RSTIB-MLP has better, or comparably good, robustness, while achieveing substantially improved computationally efficiency.

### K.7 PERFORMANCE COMPARISON WITH TRANSFORMER-BASED BASELINES

In this section, we aim to investigate the performance comparison with large amount of parameters equipped transformer based baselines with RSTIB-MLPs. **PDFormer** (Jiang et al., 2023a), **STAEformer** (Liu et al., 2023) are chosen as the baseline models, which are designed for spatial-temporal traffic forecasting, thus the results for the PEMS08 and PEMS04 datasets are provided in Table 18 and Fig.12.

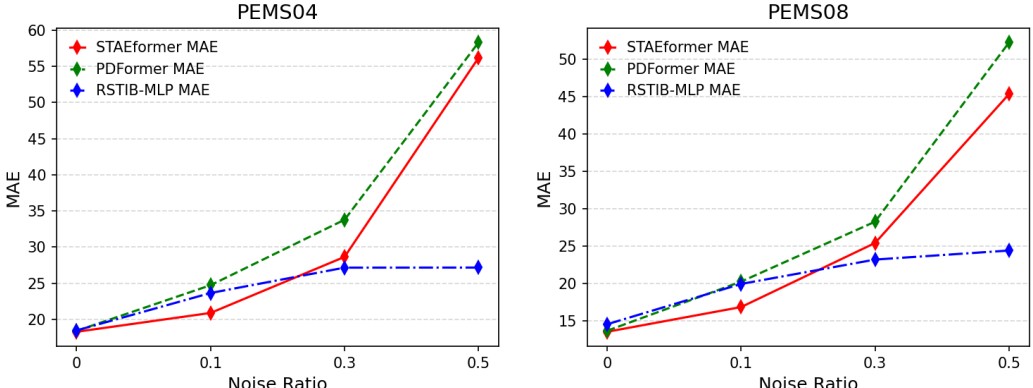

**Figure 12: MAE metric of different baselines on PEMS04 and PEMS08 when Subjecting to Noises**

As expected, it is observed that RSTIB-MLP is less effective under cleaner conditions, while it shows superior performance as the noise ratio increases. This suggests that STAEformer and PDFormer may experience faster degradation in performance under noisy conditions due to their complex architectures, which is also pointed out in (Yi et al., 2024). Thus, this analysis aids in understanding the trade-off between efficiency and robustness against noise when selecting models for practical deployment.

**Table 18: Performance Comparison with Transformer-based Baselines**

| Noise Ratio | 0%(clean) | | | 10% | | | 30% | | | 50% | | |
|---|---|---|---|---|---|---|---|---|---|---|---|---|
| Metrics | MAE | RMSE | MAPE | MAE | RMSE | MAPE | MAE | RMSE | MAPE | MAE | RMSE | MAPE |
| Dataset | PEMS08 | | | | | | | | | | | |
| STAEformer | **13.50** | **23.11** | **8.96%** | **16.81** | **26.09** | **14.89%** | 25.38 | 42.26 | 15.62% | 45.30 | 63.89 | 32.69% |
| PDFormer | 13.64 | 23.54 | 9.09% | 20.21 | 32.35 | 15.52% | 28.23 | 43.22 | 17.84% | 52.20 | 67.12 | 35.26% |
| RSTIB-MLP | 14.51 | 24.18 | 9.44% | 19.90 | 31.86 | 12.92% | **23.16** | **40.46** | **14.26%** | **24.37** | **43.77** | **14.36%** |
| Dataset | PEMS04 | | | | | | | | | | | |
| STAEformer | **18.27** | 30.38 | 12.10% | **20.88** | **32.05** | **14.02%** | 28.64 | 43.84 | 19.10% | 56.20 | 74.20 | 38.02% |
| PDFormer | 18.40 | **29.94%** | **12.04%** | 24.72 | 38.25 | 16.31% | 33.78 | 45.21 | 21.93% | 58.32 | 76.23 | 39.45% |
| RSTIB-MLP | 18.46 | 30.14 | 12.22% | 23.64 | 36.44 | 15.22% | **27.15** | **42.85** | **17.19%** | **27.16** | **43.43** | **17.76%** |

### K.8 MODEL INTERPRETATION CASE STUDY

To gain deeper insights into the learned intermediate representations, we tend to visualize the representations learnt by different models. Specifically, GWN (Wu et al., 2019), STID (Shao et al., 2022a), STGKD (Tang et al., 2024), RSTIB-MLP are included as the case models. The case study we conduct follows the steps below:

First, representations of the spatial-temporal signals in the test set are mapped into a $\mathcal{R}^2$ space using t-SNE method for dimension reduction. Then, Gaussian Kernel Density Estimation is adopted to estimate the distribution of the embeddings. The models are all trained under noise perturbation with the noise ratio = 0.1. We can tell from the results that baselines except RSTIB-MLP tend to result in the fragmentation of regions into several disconnected subspaces, or collapse into just individual region. In comparison to this, RSTIB-MLP can be more effective in organizing different spatial regions into larger subspaces with a better cohesion.

### K.9 FULL HYPERPARAMETER INVESTIGATION RESULTS

We have undertaken a hyperparameter investigation, where we selectively vary specific hyperparameters while maintaining the rest at their default settings. Our investigation centers on 2 kinds of key hyperparameters: the Lagrange multipliers in RSTIB-MLP, denoted as $\lambda(\lambda_x, \lambda_y, \lambda_z)$ and the distance function to calculate the noise

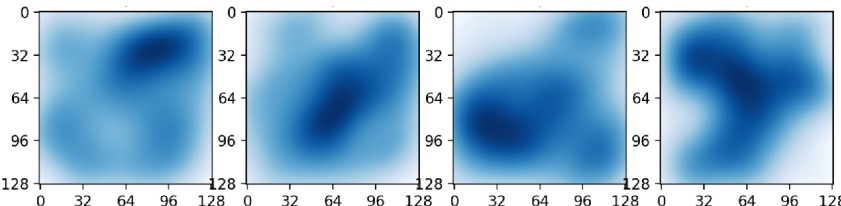

**Figure 13: Model Interpretation Case Study: Representations of the Spatial-temporal signals in the test set are mapped into a $\mathcal{R}^2$ space using t-SNE method for dimension reduction. Then, Gaussian Kernel Density Estimation is adopted to estimate the distribution of the embeddings.(Learned by GWN, STID, STGKD, RSTIB-MLP from the left to right respectively, under the noisy PEMS04 dataset with noise ratio = 30%)**

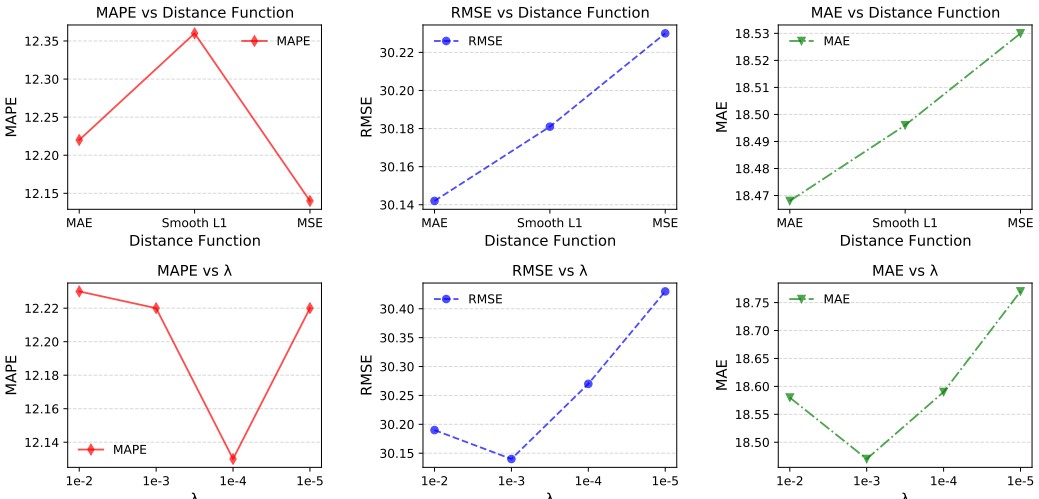

**Figure 14: Hyperparameter Analysis of $\lambda$ and Distance Function**

impact indicator $\hat{\alpha}$. We are conducting this comprehensive study to understand how these hyperparameters influence the overall model performance. The experimental outcomes on the PEMS04 dataset are presented in Fig. 14. Here follows what we have drawn from our observations: i) **The Lagrange Multipliers** We set $\lambda_x, \lambda_y, \lambda_z$ to be the same with each other and vary within the range of $1 \times 10^{-2}, 1 \times 10^{-3}, 1 \times 10^{-4}, 1 \times 10^{-5}$. ii) **Distance function to calculate noise impact indicators**. We explore different distance functions to calculate the impact indicator for knowledge distillation. Our options for the distance functions include Mean Absolute Error(MAE), Mean Squared Error(MSE), and Smooth L1 Loss(SmoothL1).

We observe that, in the PEMS04 dataset, concerning the choice of $\lambda$, setting $\lambda = 1 \times 10^{-3}$ allows the MAE and RMSE values to achieve the best results. In contrast, setting $\lambda = 1 \times 10^{-4}$ yields an optimal value for MAPE. As for selecting the distance function, using MAE as the distance function leads to the best outcomes for the corresponding MAE and RMSE metrics. Meanwhile, employing the MSE to compute the impact indicator results in optimized MAPE.

**Table 19: Teacher Model Agnostic on PEMS07 Dataset with Varied Noise Ratios**

| Noise Ratio($\gamma$) | 0% | | | 50% | | | 90% | | |
|---|---|---|---|---|---|---|---|---|---|
| | MAE | RMSE | MAPE(%) | MAE | RMSE | MAPE(%) | MAE | RMSE | MAPE(%) |
| STID | 20.41 | 33.68 | 8.74 | 32.38 | 57.29 | 14.07 | 33.33 | 58.50 | 13.96 |
| STGKD | 20.30 | 34.30 | 8.87 | 32.16 | 56.89 | 14.08 | 34.59 | 59.24 | 14.24 |
| STExplainer-CGIB | 20.55 | 35.12 | 8.61 | 35.12 | 59.17 | 16.78 | 44.14 | 69.23 | 18.87 |
| STExplainer | 20.00 | 33.45 | 8.51 | 32.52 | 57.64 | 15.48 | 45.37 | 70.57 | 19.90 |
| TrendGCN | 20.43 | 34.32 | 8.51 | 36.78 | 57.89 | 23.22 | 55.99 | 80.22 | 32.76 |
| STG-NCDE | 20.53 | 33.84 | 8.80 | 33.48 | 58.83 | 16.78 | 46.20 | 66.33 | 21.32 |
| Ours-t-MLP | 19.93 | 34.11 | 8.36 | 30.74 | 56.02 | 12.95 | 31.36 | 57.50 | 13.08 |
| Ours-t-STGCN | 19.84 | 33.90 | 8.33 | 30.94 | 56.79 | 12.91 | 30.93 | 56.91 | 12.91 |

## K.10 TEACHER MODEL AGNOSTIC STUDY

We assert that our superior performance is independent of the choice of the teacher model. Table 19 presents our results, where *Ours-t-MLP* indicates the adoption of MLP networks as the teacher model, and *Ours-t-STGCN* indicates the adoption of STGCN networks as the teacher model. It is important to note that the teacher models are pre-trained, with their parameters fixed during the training of the RSTIB-MLP. A plausible explanation for these statistics is that we aim to obtain the normalized indicators, thus indicating a relative relationship among time series. Consequently, the overall performance of its different teacher models does not significantly influence the RSTIB-MLP's performance, allowing for a more flexible configuration.

## K.11 REPLICATION STUDY

This section provides the statistically significant robustness study of the RSTIB-MLP compared with some chosen baselines when subjected to random initialization. We conducted multiple experiments on the PEMS 04/07/08 datasets, selecting five random seeds and five noisy conditions to ensure statistical significance. We report the average performance and standard deviation. The statistical outcomes of this investigation are detailed in Table 20, Table 21, Table 22, Table 23, Table 24. The empirical findings indicate RSTIB-MLP's remarkable resilience to various initialization conditions.

**Table 20: Replication Study for Performance Comparison Under Noise Perturbation with Noise Ratio $\gamma$ = 10% on three Datasets. The boldface means the best results.**

| Dataset | Method Metric | STID | STGKD | STExplainer-CGIB | STExplainer | TrendGCN | STG-NCDE | RSTIB-MLP |
|---|---|---|---|---|---|---|---|---|
| PEMS04 | MAE | 27.79±0.45 | 24.26±0.41 | 25.86±0.41 | 24.51±0.26 | 23.76±0.15 | 25.02±0.21 | **23.70 ± 0.50** |
| | RMSE | 41.45±0.07 | 37.13±0.24 | 38.37±0.48 | 36.94±0.24 | 37.06±0.27 | 37.31±0.15 | **36.58 ± 0.44** |
| | MAPE(%) | 17.41±0.37 | 16.13±0.33 | 16.00±0.17 | 16.01±0.29 | 17.56±0.05 | 17.48±0.27 | **15.33 ± 0.22** |
| PEMS07 | MAE | 27.87±0.22 | 26.89±0.23 | 27.98±0.49 | 28.26±0.30 | 26.76±0.25 | 28.87±0.41 | **26.64 ± 0.27** |
| | RMSE | 45.10±0.45 | 43.64±0.45 | 43.89±0.35 | 44.07±0.15 | 44.70±0.03 | 44.55±0.41 | **43.82 ± 0.48** |
| | MAPE(%) | 12.33±0.42 | 12.19±0.46 | 12.00±0.29 | 12.13±0.19 | 14.61±0.01 | 14.19±0.05 | **11.55 ± 0.15** |
| PEMS08 | MAE | 20.26±0.41 | 20.78±0.41 | 23.53±0.21 | 20.22±0.41 | 20.65±0.21 | 21.29±0.25 | **19.91 ± 0.02** |
| | RMSE | 32.30±0.41 | 32.52±0.49 | 35.43±0.11 | 32.69±0.20 | 32.65±0.39 | 33.16±0.32 | **32.04 ± 0.39** |
| | MAPE(%) | 14.17±0.11 | 15.11±0.14 | 24.28±0.21 | 13.52±0.09 | 14.92±0.45 | 15.27±0.04 | **13.10 ± 0.36** |

**Table 21: Replication Study for Performance Comparison Under Noise Perturbation with Noise Ratio $\gamma$ = 30% on three Datasets. The boldface means the best results.**

| Dataset | Method Metric | STID | STGKD | STExplainer-CGIB | STExplainer | TrendGCN | STG-NCDE | RSTIB-MLP |
|---|---|---|---|---|---|---|---|---|
| PEMS04 | MAE | 36.46±0.50 | 28.64±0.09 | 31.78±0.39 | 31.41±0.49 | 27.16±0.34 | 29.10±0.11 | **27.31 ± 0.05** |
| | RMSE | 52.60±0.05 | 44.85±0.12 | 48.47±0.18 | 46.98±0.07 | 42.96±0.11 | 44.36±0.42 | **43.03 ± 0.47** |
| | MAPE(%) | 21.23±0.43 | 17.48±0.07 | 18.09±0.16 | 17.98±0.45 | 19.43±0.43 | 19.26±0.02 | **17.21 ± 0.44** |
| PEMS07 | MAE | 31.87±0.49 | 31.59±0.39 | 34.89±0.20 | 31.43±0.23 | 31.82±0.11 | 32.29±0.19 | **30.29 ± 0.15** |
| | RMSE | 55.21±0.03 | 55.24±0.29 | 57.44±0.16 | 54.21±0.09 | 55.23±0.40 | 56.29±0.04 | **54.14 ± 0.30** |
| | MAPE(%) | 13.63±0.07 | 13.62±0.33 | 14.32±0.20 | 14.98±0.27 | 20.68±0.33 | 15.84±0.39 | **12.80 ± 0.34** |
| PEMS08 | MAE | 26.61±0.42 | 25.65±0.45 | 24.88±0.38 | 25.60±0.30 | 24.90±0.42 | 28.41±0.40 | **23.24 ± 0.20** |
| | RMSE | 45.79±0.08 | 43.46±0.28 | 43.15±0.15 | 43.37±0.37 | 41.58±0.40 | 41.99±0.10 | **40.47 ± 0.02** |
| | MAPE(%) | 15.64±0.35 | 16.21±0.29 | 15.44±0.21 | 16.75±0.47 | 23.55±0.38 | 16.29±0.45 | **14.38 ± 0.27** |

**Table 22: Replication Study for Performance Comparison Under Noise Perturbation with Noise Ratio $\gamma$ = 50% on three Datasets. The boldface means the best results.**

| Dataset | Method Metric | STID | STGKD | STExplainer-CGIB | STExplainer | TrendGCN | STG-NCDE | RSTIB-MLP |
|---|---|---|---|---|---|---|---|---|
| PEMS04 | MAE | 36.26±0.40 | 29.15±0.29 | 28.48±0.07 | 29.45±0.44 | 27.90±0.05 | 31.06±0.29 | **27.29 ± 0.06** |
| | RMSE | 52.32±0.21 | 46.40±0.09 | 44.74±0.05 | 46.50±0.46 | 44.81±0.15 | 47.29±0.36 | **43.60 ± 0.46** |
| | MAPE(%) | 21.42±0.01 | 18.56±0.33 | 16.88±0.30 | 17.23±0.34 | 20.31±0.39 | 20.74±0.41 | **17.60 ± 0.12** |
| PEMS07 | MAE | 32.56±0.15 | 32.10±0.28 | 35.08±0.44 | 32.53±0.49 | 36.82±0.11 | 33.34±0.39 | **30.97 ± 0.15** |
| | RMSE | 57.40±0.00 | 56.93±0.30 | 59.06±0.17 | 57.47±0.16 | 57.72±0.21 | 58.99±0.24 | **56.92 ± 0.22** |
| | MAPE(%) | 13.91±0.22 | 14.02±0.33 | 16.75±0.26 | 15.41±0.47 | 23.41±0.11 | 16.76±0.34 | **12.98 ± 0.39** |
| PEMS08 | MAE | 27.73±0.46 | 25.88±0.24 | 26.63±0.32 | 27.16±0.02 | 26.86±0.25 | 29.40±0.01 | **24.47 ± 0.24** |
| | RMSE | 48.49±0.24 | 43.94±0.16 | 44.60±0.43 | 45.70±0.03 | 45.67±0.26 | 47.13±0.25 | **43.84 ± 0.26** |
| | MAPE(%) | 16.27±0.26 | 16.77±0.37 | 15.61±0.17 | 15.30±0.40 | 23.15±0.44 | 18.45±0.47 | **14.41 ± 0.24** |

**Table 23: Replication Study for Performance Comparison Under Noise Perturbation with Noise Ratio $\gamma$ = 70% on three Datasets. The boldface means the best results.**

| Dataset | Method Metric | STID | STGKD | STExplainer-CGIB | STExplainer | TrendGCN | STG-NCDE | RSTIB-MLP |
|---|---|---|---|---|---|---|---|---|
| PEMS04 | MAE | 31.34±0.47 | 29.86±0.48 | 38.56±0.38 | 33.31±0.38 | 33.07±0.34 | 33.21±0.09 | **27.08 ± 0.26** |
| | RMSE | 47.12±0.28 | 46.06±0.30 | 58.65±0.24 | 52.14±0.17 | 51.04±0.42 | 49.79±0.27 | **43.12 ± 0.22** |
| | MAPE(%) | 18.49±0.17 | 17.66±0.04 | 21.69±0.27 | 19.26±0.25 | 24.38±0.02 | 22.24±0.37 | **17.42 ± 0.12** |
| PEMS07 | MAE | 32.42±0.03 | 32.80±0.18 | 41.48±0.41 | 43.83±0.09 | 43.07±0.05 | 43.25±0.32 | **31.02 ± 0.25** |
| | RMSE | 58.15±0.28 | 57.58±0.41 | 66.23±0.22 | 68.77±0.48 | 64.76±0.18 | 63.26±0.29 | **57.03 ± 0.20** |
| | MAPE(%) | 13.89±0.47 | 14.17±0.22 | 17.30±0.23 | 17.95±0.38 | 29.23±0.28 | 18.76±0.28 | **13.01 ± 0.20** |
| PEMS08 | MAE | 26.32±0.00 | 25.02±0.08 | 28.28±0.09 | 28.15±0.49 | 32.49±0.40 | 30.35±0.15 | **24.36 ± 0.30** |
| | RMSE | 45.72±0.22 | 45.55±0.36 | 45.83±0.02 | 46.96±0.24 | 47.82±0.23 | 49.25±0.22 | **43.70 ± 0.50** |
| | MAPE(%) | 16.54±0.18 | 15.04±0.10 | 17.55±0.25 | 16.86±0.07 | 28.25±0.20 | 20.15±0.00 | **14.24 ± 0.28** |

**Table 24: Replication Study for Performance Comparison Under Noise Perturbation with Noise Ratio $\gamma$ = 90% on three Datasets. The boldface means the best results.**

| Dataset | Method Metric | STID | STGKD | STExplainer-CGIB | STExplainer | TrendGCN | STG-NCDE | RSTIB-MLP |
|---|---|---|---|---|---|---|---|---|
| PEMS04 | MAE | 33.62±0.06 | 29.26±0.10 | 33.11±0.39 | 34.37±0.44 | 46.22±0.12 | 37.06±0.42 | **28.11 ± 0.29** |
| | RMSE | 49.02±0.41 | 45.89±0.37 | 49.52±0.09 | 53.38±0.26 | 67.66±0.03 | 56.11±0.40 | **44.67 ± 0.41** |
| | MAPE(%) | 20.37±0.26 | 17.88±0.21 | 24.49±0.46 | 26.70±0.41 | 38.52±0.27 | 25.79±0.24 | **17.05 ± 0.36** |
| PEMS07 | MAE | 33.42±0.05 | 34.77±0.39 | 44.25±0.13 | 45.40±0.49 | 55.86±0.09 | 46.33±0.07 | **31.00 ± 0.43** |
| | RMSE | 58.39±0.12 | 59.33±0.10 | 69.36±0.29 | 70.63±0.43 | 80.23±0.41 | 66.36±0.40 | **56.96 ± 0.16** |
| | MAPE(%) | 14.13±0.32 | 14.09±0.32 | 19.04±0.14 | 19.95±0.17 | 32.67±0.05 | 21.25±0.40 | **13.10 ± 0.06** |
| PEMS08 | MAE | 26.50±0.27 | 25.37±0.46 | 32.71±0.02 | 28.72±0.17 | 46.71±0.48 | 32.59±0.04 | **24.43 ± 0.30** |
| | RMSE | 45.26±0.13 | 44.89±0.07 | 53.49±0.25 | 47.20±0.18 | 65.12±0.26 | 53.41±0.33 | **44.04 ± 0.39** |
| | MAPE(%) | 15.97±0.03 | 15.64±0.37 | 20.04±0.05 | 18.27±0.48 | 38.77±0.39 | 23.44±0.12 | **14.22 ± 0.47** |

