# OpenReview forum: "Combating Dual Noise Effect in Spatial-temporal Forecasting via Information Bottleneck Principle"
_ICLR.cc/2025/Conference — Submitted to ICLR 2025_

### Official Review · Reviewer_6Qyp · 2024-11-02

**Soundness:** 2
**Presentation:** 3
**Contribution:** 2
**Rating:** 3
**Confidence:** 3

**Summary:**

This work propose the Robust Spatial-Temporal Information Bottleneck (RSTIB) principle for guiding robust representation learning to mitigate these effects. By leveraging RSTIB, the authors instantiate method using a pure MLP network, resulting in a computationally efficient and robust RSTIB-MLP model for the task.

**Strengths:**

**S1.** This work is interesting, and has a strong theoretical foundation.

**S2.** The experiments are detailed and comprehensive.

**Weaknesses:**

**W1.** I'm confused about your research motivations. In Sec.1, the manuscript describes many issues, including efficiency problem, the sample indistinguishability issue, and the problem of feature collapse, which seem to have very shallow correlations. Therefore, I'm unclear which of these challenges inspire your research, making it difficult for me to understand the purpose of this work.

**W2.** The manuscript is poor-writen. There is a lack of clear explanations regarding the transitions from the Markov assumption  $Y - X - Z$ to  $Z - X - Y$, and then to $X - Z - Y$, as well as an analysis of their differences. For example, both $Y - X - Z$  and $Z - X - Y$  indicate that $Z$  and  $Y$  are conditionally independent of $X$; what are the significant differences between them?


**W3.** In Sec. 4.1, much of the analysis builds on existing works, such as [Wieczorek & Roth], [Alemi et al.], and [Lim & Puthusserypady]. What is your innovative contributions? I guess your novel insight lies in the proposed $Z - X - Y$ assumption, but Definition 4.1 is based on the Markov chain condition  $X - Z - Y$. Intuitively, while the authors present a lot of content and information, it does not provide readers with a sense of rich and novel insights; rather, it comes across as confusing.

**W4.** It would be beneficial to add an overview framework figure in the manuscript to help readers better access the model you proposed.

**W5.** Sec.7 exceeds the page limit (maximum 10 pages) of submission rules. I will submit this issue to PCs.

**Questions:**

**Q1**. I would like to know if the use of MLP in the instantiation process of RSTIB is irreplaceable. In the discussion of Sec. 4.2, I do not see the necessity of MLP in the instantiation of the RSTIB process. It seems that other network modules could achieve similar goals.

**Q2**. I'm a confused about the details of the dataset you selected. Which year does the SD dataset you used come from?

---

> ### Comment · Reviewer_6Qyp · 2024-11-24
> **Replay**
>
> Thank you very much for your detailed reply, but I still have the following concerns that you need to clarify.
>
> Q1.        I remain confused about the authors' motivation.        They claim to address the noise learning problem in MLPs - why emphasize only MLPs?        Don't GCNs face similar issues?
>
> Q2.     The authors have not provided detailed explanations about noise addition. Moreover, a crucial issue is that the noise they focus on appears to be artificially added. Since such pristine conditions do not exist in reality, why are we engineering a research problem? This artificial dataset creation through noise addition is unnatural and lacks practical significance—it alters the data distribution and disrupts original time series properties such as lag and periodicity. I believe that performance comparisons on such datasets lack strong motivation. However, if we disregard the controversial artificial noise addition and evaluate only on datasets with real-world noise, the performance improvements reported in Table 2 are extremely limited. In many cases, especially in the 0% (clean) portion of original real datasets, the relative improvements are less than 2%, which may fall within the margin of learning error and could potentially be compensated through parameter tuning, or may arise from the additional parameter layers introduced by the authors, or other enhancement strategies. Therefore, I remain confused about the practical significance and motivation behind RSTIB-MLP.
>
> Q3.        I suggest including more advanced baselines, such as STD-MAE, STEP, STAEformer, D²STGNN, and time series models like TimeMixer, PatchMixer, PathTST, which perform well on Electricity data.
>
> Q4.        The authors should conduct experiments on larger datasets to demonstrate their claimed speed advantages, rather than just using a few hundred nodes.       And why wasn't STID efficiency compared in the tables?
>
> Q5.        While the introduction emphasizes addressing noise learning in MLP-based spatiotemporal models, can the proposed model be applied to other GCN/Transformer models?        To my knowledge, the latter currently dominate this field, which limits the paper's broader applicability.
>
> Q6.        I believe RSTIB-MLP's performance largely stems from STID's enhanced embedding of time of day and day of week, along with spatial node-embedding.        The authors should conduct more comprehensive ablation studies on RSTIB-MLP's embedding to highlight the importance of other modules.
>
> Q7.        The authors claim in the abstract that "the representation learning guided by RSTIB can be more robust against noise interference."        To prove this, reporting model performance under different noise ratios is insufficient.        They should also report the relative performance loss as noise increases.        Lower relative performance loss compared to baselines would demonstrate RSTIB-MLP's ability to overcome noise effects.        Based on this, I find that RSTIB-MLP's performance degradation isn't the best - for instance, according to PEMS04 data in Table 2, when calculating MAE decline ratios for each noise level relative to the previous level, TrendGCN shows smaller performance loss than RSTIB-MLP, despite RSTIB-MLP's superior absolute performance.        This phenomenon doesn't adequately support the claim that "the representation learning guided by RSTIB can be more robust against noise interference."        I hope the authors can provide more comprehensive reporting and explanation of such data.
>
> Q8.     The experimental details are seriously lacking.     Was the data noise added before or after data normalization?     I observed that the performance of STID drops rapidly as the proportion of noise increases.     Is this because the author also unnecessarily added noise to the index information about time of day and day of week (usually corresponding to the second and third features of the traffic dataset) when adding noise to the data? The proposed final loss involves multiple loss balancing factors. I think a detailed hyperparameter evaluation is absolutely necessary, and providing a deeper discussion can make the model clearer.
>
> Q9. The authors state that "Spatial-temporal prompts (Tang et al., 2024) are utilized to attach the spatial-temporal information".  Is this an essential component?  Were similar prompts added to other comparison models?  Does this potentially involve unfair comparisons?  Have ablation experiments been conducted regarding these prompts?

---

### Official Review · Reviewer_wn2Q · 2024-11-02

**Soundness:** 2
**Presentation:** 3
**Contribution:** 2
**Rating:** 6
**Confidence:** 3

**Summary:**

This paper introduces RSTIB (Robust Spatial-Temporal Information Bottleneck), a method to enhance the robustness of spatio-temporal forecasting models in noisy environments by explicitly minimizing noise information through new information terms. The key strengths of the paper include a strong theoretical foundation, practical implementation, and empirical evidence showing that RSTIB-MLP outperforms existing methods on various benchmark datasets. However, the experiments primarily use artificially added noise, which may not fully represent the complexity and variability of real-world noise. Additionally, the validation process is limited to Multi-Layer Perceptron (MLP)-based approaches, restricting the understanding of RSTIB's performance in different model architectures.

**Strengths:**

- **Theoretically Extends Information Bottleneck (IB)**: RSTIB introduces a new principle that extends the traditional IB to handle dual noise effects in spatial-temporal forecasting. This extends the IB method to better address the complexity and dynamics of such data.

- **Empirical Evidence**: Extensive experiments on multiple benchmark datasets show that RSTIB-MLP outperforms existing methods, especially under noisy conditions.

- **Clarity**: The article is well-structured, with clear and concise explanations of equations and figures. The explanations of the theoretical background and experimental setup are easy to follow.

**Weaknesses:**

- **Limited Model Scope**: The primary modification to RSTIB is the loss function, and the validation process is confined to Multi-Layer Perceptron (MLP)-based approaches. This limits our understanding of RSTIB's performance when applied to different models.

- **Insufficient Baseline Comparison**: While RSTIB demonstrates strong performance on noisy datasets, the lack of direct comparison with baselines using real-world clean data can lead to misinterpretation of the performance gains. These improvements might be attributed to data preprocessing rather than the method itself.

- **Limitations of Artificial Noise**: The experiments use artificially added noise to evaluate the model. While this provides controlled conditions, it does not fully capture the diversity and complexity of real-world noise, potentially leading to an incomplete assessment of RSTIB's robustness.

**Questions:**

- Can you extend RSTIB's validation to include comparisons with other spatiotemporal baselines? This would help show RSTIB's broader applicability.

- Can you conduct additional experiments using different types or definitions of real-world noise, such as varying levels of missing data, sensor malfunctions, or environmental disturbances? This would give more insight into RSTIB's robustness and effectiveness in practical scenarios.

- Since real-world data is important, can you perform an ablation study on clean datasets to understand how different components of RSTIB-MLP contribute to its performance? Specifically, how do individual components affect the overall performance in real-world scenarios?

- To validate RSTIB more rigorously, can you perform an ablation study on different models (e.g., STGKD, STGCN) using clean and noisy datasets? This would help clarify how RSTIB performs across various model architectures and ensure the improvements are consistent and robust.

---

> ### Comment · Reviewer_wn2Q · 2024-11-26
>
> Thank you for your prompt response.
>
> Regarding Q1 and Q4, I share a similar inquiry with Reviewer 6Qyp: Is the use of MLP in the instantiation of RSTIB indispensable, or could it be substituted with another component without compromising performance? It would be great if you could provide further experimental outcomes, such as principal performance comparisons and ablation studies, illustrating how RSTIB functions alongside various base models. These additional findings will be referenced in the final version of the manuscript. As for Q2 and Q3, I have no further concerns.

---

> ### Comment · Reviewer_wn2Q · 2024-12-01
>
> Thanks for the detailed responses. RSTIB is indeed an interesting research direction. However, its applicability appears limited, as its strong performance is primarily demonstrated on MLPs, with less notable results on other STGNNs. This may restrict the generalizability of the proposed architecture.
>
> I appreciate the authors’ efforts to address most of my concerns and have updated my rating accordingly.

---

> ### Author Response · Authors · 2024-12-01
>
> Thank you for the prompt feedback regarding the results.  As a general principle designed for combating dual noise effect, RSTIB has good applicability. Its instantiation RSTIB-MLP shows strong robustness while enjoining the efficiency of MLPs, but indeed its extension to other models like STGNNs requires further investigation. Thank you very much again for the great efforts you have devoted to review our submission and your positive support.

---

### Official Review · Reviewer_b2h4 · 2024-11-03

**Soundness:** 3
**Presentation:** 3
**Contribution:** 3
**Rating:** 5
**Confidence:** 4

**Summary:**

The paper introduces the Robust Spatial-Temporal Information Bottleneck (RSTIB) principle for spatial-temporal forecasting. This method addresses issues in existing MLP-based models, especially their susceptibility to dual noise effect, where noise impacts both input and target data in spatial-temporal forecasting. RSTIB adapts the Information Bottleneck principle to mitigate this effect, enhancing robustness by minimizing irrelevant noise without relying on the conventional Markov assumption. The proposed RSTIB-MLP model, leveraging MLPs, achieves efficient and resilient forecasting, outperforming state-of-the-art Spatial-Temporal Graph Neural Networks (STGNNs) in accuracy and speed on six benchmark datasets. Additionally, a knowledge distillation module further boosts robustness by preserving feature diversity and minimizing feature collapse in noisy conditions.

**Strengths:**

This paper introduces the novel RSTIB framework to handle dual noise (affecting both input and target data) in spatial-temporal forecasting. By extending the IB principle without the Markov assumption, and integrating MLP-based modeling for efficiency, the authors offer an interesting adaptation of established ideas to tackle a key limitation in noisy spatial-temporal forecasting.

Extensive experiments on six benchmark datasets were conducted, showing consistent improvements in forecasting accuracy and efficiency over existing models. The authors support their claims with both theoretical analysis and an ablation study, demonstrating the robustness and efficiency of RSTIB compared to state-of-the-art models.

The paper clearly explains the dual noise challenge and the motivation behind RSTIB. While some theoretical sections are complex, the structure and use of diagrams make the overall approach understandable. Simplifying certain explanations could further improve accessibility.

**Weaknesses:**

The role of knowledge distillation in enhancing feature diversity is not fully explored. While it is included in the training regime, a more detailed explanation of its integration into RSTIB would strengthen understanding. The paper could improve by clarifying the impact of knowledge distillation on feature variance. Perhaps explaining why and how using KD show a small impact on prediction performance in Fig. 4 but large impact on feature variance Fig. 3 would help.

The ablation study could further analyze the impact of each regularization term (e.g., input, target and representation regularizations in RSTIB). A finer-grained analysis of these terms would clarify their individual contributions to robustness and may offer insights for optimizing the framework.

While RSTIB-MLP demonstrates computational efficiency per training epoch, reporting the total training time to convergence would provide a clearer picture of its efficiency. Since different algorithms may require varying numbers of epochs to converge, total training time would offer a more comprehensive comparison of computational performance across models.

**Questions:**

1. The knowledge distillation module is mentioned as enhancing feature diversity, but its exact role in balancing informative terms is somewhat ambiguous. Could the authors elaborate on how knowledge distillation specifically influences feature variance and robustness?

2. Would the authors consider an ablation study that examines each regularization term within the RSTIB objective (input, target, and representation regularizations) separately?

3. Although the paper reports computational efficiency per epoch, would the authors consider including total training time until convergence for each method?

---

### Official Review · Reviewer_JpFa · 2024-11-04

**Soundness:** 3
**Presentation:** 3
**Contribution:** 2
**Rating:** 6
**Confidence:** 5

**Summary:**

The authors propose the Robust Spatial-Temporal Information Bottleneck (RSTIB) principle, which extends previous Information Bottleneck (IB) approaches by lifting the specific Markov assumption without impairing the IB nature. Then, by explicitly minimizing the irrelevant noisy information, the representation learning guided by RSTIB can be more robust against noise interference. The extensive experimental results on six intrinsically noisy benchmark datasets from various domains show that the RSTIB-MLP runs much faster than state-of-the-art STGNNs and delivers superior forecasting accuracy across noisy environments, substantiating its robustness and efficiency.

**Strengths:**

1. Studying the effect of noise on spatio-temporal prediction is a novel topic, which will bring new insights to the field of spatio-temporal prediction.

2. In the experimental evaluation section, the baselines and datasets used were comprehensive, especially some large datasets were used.

3. The effectiveness of the proposed method is further verified by a comparative experiment of efficiency.

**Weaknesses:**

1. The proposed methods appear to have limited performance gains, such as those in the LargeST(SD) and Weather2K-R datasets in Table 2.

2. The authors don't seem to have made their source code available to directly verify availability and reproducibility.

3. Some important hyperparameter experiments should be included in the experimental section of the main text.

**Questions:**

See Weaknesses.

---

### Official Review · Reviewer_nqwf · 2024-11-05

**Soundness:** 3
**Presentation:** 2
**Contribution:** 2
**Rating:** 6
**Confidence:** 3

**Summary:**

This paper presents a novel approach to enhance the robustness of spatial-temporal forecasting models against noise interference. The authors propose the Robust Spatial-Temporal Information Bottleneck (RSTIB) principle, which extends previous Information Bottleneck (IB) methods by lifting the specific Markov assumption without compromising the IB nature.

**Strengths:**

- The RSTIB principle is grounded in information theory, providing a solid theoretical basis for the model's design.
- The proposed RSTIB-MLP achieve SOTA forecasting performance while requires less computational resource.
- The paper provides extensive experimental results on multiple benchmark datasets, validating the effectiveness of the proposed method.

**Weaknesses:**

- The paper is based on assumption the the noise in data is Additive White Gaussian Noise (AWGN). I wonder whether the derivation holds if the noise type is not AWGN.
- In Sec4.2, it seems that MLP is not a necessity of implementing RSTIB, what if we utilize other network modules?
- Authors should emphasize the differences between this work and existing literatures, since much of the analysis in Sec4.1 is building upon existing works, including [1,2,3].

[1] Alexander A Alemi, Ian Fischer, Joshua V Dillon, and Kevin Murphy. Deep variational information bottleneck. arXiv preprint arXiv:1612.00410, 2016.
[2] Aleksander Wieczorek and Volker Roth. On the difference between the information bottleneck and
the deep information bottleneck. Entropy, 22(2):131, 2020.
[3] Teck Por Lim and Sadasivan Puthusserypady. Chaotic time series prediction and additive white gaussian noise. Physics letters A, 365(4):309–314, 2007.

**Questions:**

See Weaknesses.

---

### Meta-Review · Area_Chair_Aovr · 2024-12-19

**Metareview:**

This paper proposes an approach by extending the information bottleneck principle to enhance the robustness of spatial-temporal forecasting models against noise interference.

Major strengths:
- The information-theoretic RSTIB principle is mathematically justified.
- The proposed RSTIB-MLP shows promises when compared with other methods in the reported experiments.

Major weaknesses:
- The improvement is not always significant.
- The effectiveness of the principle on other model types is unclear.

We appreciate the effort of the authors to engage in discussions with the reviewers and conducting additional experiments to address some of the concerns or questions raised. Despite its merits, the effectiveness of the proposed RSTIB principle needs to be justified more convincingly due partially to the weaknesses listed above. This is probably the main reason why some reviewers still hold a negative view and those who are positive are unwilling to cast a stronger vote to champion this paper for acceptance. The paper will be more ready for publication if a more thorough study is reported by including more model types, among other things. The authors are encouraged to improve their paper for future submission by considering the comments and suggestions of the reviewers.

**Additional Comments On Reviewer Discussion:**

The authors engaged in detailed discussions with some reviewers and provided further experiment results. However, they also admitted that they were not able to expand the study to include more model types during the discussion period.

---

### Decision · Program_Chairs · 2025-01-22

Reject